

# The control of hydrogen sulfide on benthic iron and cadmium fluxes in the oxygen minimum zone off Peru

Anna Plass[1][*], Christian Schlosser[1], Stefan Sommer[1], Andrew W. Dale[1], Eric P. Achterberg[1], Florian Scholz[1][*]

[1]GEOMAR Helmholtz Centre for Ocean Research Kiel, Wischhofstraße 1-3, 24148 Kiel, Germany

[*]Correspondence to: Anna Plass (aplass@geomar.de), Florian Scholz (fscholz@geomar.de)



## Abstract

Sediments in oxygen-depleted marine environments can be an important sink or source of bio-essential trace metals in the ocean. However, the key mechanisms controlling the release from or burial of trace metals in sediments are not exactly understood. Here, we investigate the benthic biogeochemical cycling of Fe and Cd in the oxygen minimum zone off Peru. We combine bottom water profiles, pore water profiles, as well as benthic fluxes determined from pore water profiles and in-situ from benthic chamber incubations along a depth transect at 12° S. In agreement with previous studies, both concentration-depth profiles and in-situ benthic fluxes indicate a Fe release from sediments into bottom waters. Diffusive Fe fluxes and Fe fluxes from benthic chamber incubations are roughly consistent ($0.3 - 17.1$ mmol m$^{-2}$ y$^{-1}$), indicating that diffusion is the main transport mechanism of dissolved Fe across the sediment-water interface. The occurrence of mats of sulfur oxidizing bacteria on the seafloor represents an important control on the spatial distribution of Fe fluxes by regulating hydrogen sulfide (H$_2$S) concentrations and, potentially, Fe sulfide precipitation within the surface sediment. Removal of dissolved Fe after its release to anoxic bottom waters is rapid in the first 4 m away from the seafloor (half-life < 3 min) which hints to oxidative removal by nitrite or interaction with particles in the benthic boundary layer. Benthic flux estimates of Cd are indicative of a flux into the sediment within the oxygen minimum zone. Fluxes from benthic chamber incubations (up to 22.6 µmol m$^{-2}$ y$^{-1}$) exceed the diffusive fluxes (< 1 µmol m$^{-2}$ y$^{-1}$) by a factor > 25, indicating that downward diffusion of Cd across the sediment-water interface is of subordinate importance for Cd removal from benthic chambers. As Cd removal in benthic chambers co-varies with H$_2$S concentrations in the pore water of surface sediments, we argue that Cd removal is mediated by precipitation of CdS within the chamber. A mass balance approach, taking into account the contributions of diffusive fluxes and fluxes measured in benthic chambers as well as Cd delivery with organic material suggests that CdS precipitation in the near-bottom water could make an important contribution to the overall Cd mass accumulation in the sediment solid phase. According to our results, the solubility of trace metal sulfide minerals (Cd << Fe) is a key-factor controlling trace metal removal and consequently the magnitude as well as the temporal and spatial heterogeneity of sedimentary fluxes. We argue that depending on their sulfide solubility, sedimentary source or sink fluxes of trace metals will change differentially as a result of declining oxygen concentrations and an associated



expansion of sulfidic surface sediments. Such a trend could cause a change in the
trace metal stoichiometry of upwelling water masses with potential consequences for
marine ecosystems in the surface ocean.
**1. Introduction**
**1.1 Scientific rationale**
The world´s oceans are losing oxygen (e.g. Keeling et al. 2010; Stramma et al.
2010; Helm et al. 2011). In total around 2 % of oxygen has been lost over the past five
decades (Schmidtko et al., 2017) and an expansion of oxygen minimum zones (OMZs)
in the tropical oceans has been documented over the same timespan (Stramma et al.,
2008). The biogeochemical cycling of several nutrient type trace metals (TMs) is likely
to be particularly susceptible to changing oxygen concentrations as they occur in
different oxidation states (e.g. Fe, Mn, Co) and/or are precipitated as sulfide mineral in
anoxic-sulfidic environments (e.g. Zn, Cd). However, with the exception of Fe (Dale et
al., 2015a; Lohan and Bruland, 2008; Rapp et al., 2018; Schlosser et al., 2018; Scholz
et al., 2014a), little information is available on how other TM fluxes will respond to
ocean deoxygenation. As certain TMs are essential for the growth of marine organisms
(e.g. Fe, Mn, Co, Ni, Zn, Cd), TM availability can (co-)limit primary productivity and
therefore affect oceanic carbon sequestration through the biological pump (Saito et al.,
2008; Moore et al., 2013; Morel et al., 2014). As a consequence, a better
understanding of how TMs respond to low oxygen conditions is essential for predicting
how marine ecosystems and the carbon cycle will evolve in the future ocean, with
modelling scenarios predicting a continuation of ocean deoxygenation (Bopp et al.,
2002; Oschlies et al., 2008; Keeling et al., 2010)
Marine sediments are an important source or sink of TMs to the ocean under
low oxygen conditions (Böning et al., 2004; Brumsack, 2006; Scor Working Group,
2007; Severmann et al., 2010; Noble et al., 2012; Biller and Bruland, 2013; Conway
and John, 2015b; Klar et al., 2018). In the OMZ off the coast of Peru, substantial fluxes
of reduced Fe and other TMs across the sediment-bottom water interface have been





documented (Noffke et al., 2012; Scholz et al., 2016) or inferred (Hawco et al., 2016).
While a number of studies have addressed biogeochemical processes that lead to
benthic Fe release, the key biogeochemical processes and conditions that control the
sedimentary release or burial of other TMs are still poorly constrained. Moreover, a
detailed picture of removal or stabilization processes and rates that take place in the
highly dynamic benthic boundary layer is lacking.
In this article, we compare the benthic biogeochemical cycling of Fe and Cd. It
has been established that the Peruvian OMZ represents a source of dissolved Fe to
the ocean (Noffke et al., 2012; Fitzsimmons et al., 2016; John et al., 2018). In contrast,
earlier studies have demonstrated that OMZs represent a sink for Cd (Janssen et al.,
2014; Böning et al., 2004). Because of their contrasting tendency to form sulfide
minerals and different supply pathways to the sediment, Fe and Cd can provide
information about how sedimentary fluxes of different TMs may respond to declining
oxygen concentrations. By comparing the benthic biogeochemical cycling of Fe and
Cd across spatial and temporal redox gradients, we aim to provide general constraints
on how the stoichiometry of bio-essential TMs in seawater may be affected by ocean
deoxygenation.

## 1.2. Marine biogeochemistry of iron

Iron is the most abundant TM in phytoplankton and part of a range of
metalloenzymes that are involved in important biological functions, such as
photosynthesis or nitrogen fixation (Twining and Baines, 2013). Despite Fe being
highly abundant in the continental crust, its low availability limits primary productivity in
up to 30 % of the surface ocean area (Moore et al., 2013). This limitation arises from
the low solubility of its thermodynamically stable form in oxic waters, Fe(III).
Concentrations can reach up to ~ 1 nM when Fe(III) is kept in solution through
complexation with organic ligands (Rue and Bruland, 1997; Liu and Millero, 2002; Boyd
and Ellwood, 2010; Raiswell and Canfield, 2012). The thermodynamically stable form
of Fe under anoxic conditions, Fe(II), is more soluble and therefore anoxic waters are
typically characterized by higher dissolved Fe concentrations (up to tens of nM)
(Conway and John, 2014; Vedamati et al., 2014; Fitzsimmons et al., 2016; Schlosser
et al., 2018).



Sediments within OMZs are considered an important source of dissolved Fe and
some of the highest sedimentary Fe fluxes have been observed in these regions
(Severmann et al., 2010; Noffke et al., 2012). Under anoxic conditions, Fe(II) can be
liberated from the sediments into pore waters from Fe-(oxyhydr)oxides through
reductive dissolution by microbes or abiotic reduction with $H_2S$ (Canfield, 1989). In the
absence of oxygen, dissolved Fe(II) escapes the rapid re-oxidation and subsequent
(oxyhydr)oxide precipitation and can, therefore, diffuse from pore waters into bottom
waters. However, in anoxic OMZs, where denitrification takes place, Fe(II) can also be
re-oxidized by either biologically by nitrate-reducing microbes or abiotically by nitrite
(Straub et al., 1996; Carlson et al., 2013; Scholz et al., 2016; Heller et al., 2017). The
solubility of Fe in sulfidic (i.e. $NO_3^-$ and $NO_2^-$ depleted) water is relatively high (Rickard
et al., 2006) and it has been observed, that during sulfidic events dissolved Fe can
accumulate in the water column (up to hundreds of nM) because of decreased Fe
oxidation (Scholz et al., 2016) and stabilization as aqueous Fe sulfide complexes and
clusters (Schlosser et al., 2018). However, Fe fluxes across the benthic boundary have
been hypothesized to decrease as $H_2S$ accumulation in the surface sediment impedes
Fe escape through precipitation of Fe sulfide minerals (Scholz et al., 2014b).

**1.3. Marine biogeochemistry of cadmium**

Even though its concentrations are one order of magnitude lower compared to
Fe, Cd is abundant in phytoplankton (Twining and Baines, 2013). A function for Cd as
a catalytic metal atom in the carbonic anhydrase protein has been found in diatoms
(Lane and Morel, 2000) and it can also substitute Zn and enhance phytoplankton
growth under Zn limitation in different phytoplankton species (Price and Morel, 1990;
Lee and Morel, 1995; Sunda and Huntsman, 2000; Xu et al., 2008). Inside the marine
sediments Cd can be released from the solid phase into the pore waters through the
remineralization of organic matter (Klinkhammer et al., 1982; Collier and Edmond,
1984; Gendron et al., 1986; Gerringa, 1990; Audry et al., 2006; Scholz and Neumann,
2007). After its release to the pore water, Cd can either diffuse across the sediment-
water interface, or under anoxic and sulfidic conditions, Cd is thought to be precipitated
as CdS (Greenockite) (Westerlund et al., 1986; Gobeil et al., 1987; Rosenthal et al.,
1995; Audry et al., 2006). Due to its low sulfide solubility, CdS can precipitate at much





lower $H_2S$ concentrations than FeS (Mackinawite), which is the precursor for pyrite
($FeS_2$) (Morse and Luther, 1999).

142         The few studies on pore water concentration and benthic fluxes of Cd,  mostly

carried out in estuaries or coastal settings, generally concluded that the flux of organic
material and the presence of $H_2S$ are the most important factors controlling the balance
between Cd recycling versus precipitation and burial (e.g. Westerlund et al. 1986;
Colbert et al. 2001; Audry et al. 2006; Scholz and Neumann 2007). Low oxygen regions
in the ocean are considered an important sink for Cd (Janssen et al., 2014; Conway
and John, 2015a; Xie et al., 2019) and sediments below OMZs are highly enriched in
Cd (Ragueneau et al., 2000; Böning et al., 2004; Borchers et al., 2005; Muñoz et al.,
2012; Little et al., 2015), however, the respective contributions of different Cd removal
mechanism to Cd accumulation in the sediment have not been quantified.

### 1.4. Study area

154         Seasonal upwelling of nutrient-rich waters off the Peruvian coast in austral

winter leads to a high primary productivity in the euphotic zone (~ 300 mmol C $m^{-3}$ $d^{-1}$)
(Pennington et al., 2006). The combination of oxygen consumption through the
respiration of this organic matter and low oxygen concentrations in water masses that
supply upwelling regions, leads to the formation of one of the world's most intense
OMZs, with complete oxygen consumption in the core of the OMZ between ~ 100 m –
300 m water depth (Karstensen et al., 2008; Thamdrup et al., 2012). Upon oxygen
depletion, $NO_3^-$ can serve as an electron acceptor for respiration, therefore,
denitrification, dissimilatory  reduction of $NO_3^-$ to ammonium (DNRA) and anaerobic
ammonium oxidation (anammox) with $NO_2^-$ are important biogeochemical processes
within the anoxic and nitrogenous water column (Lam et al., 2009; Lam and Kuypers,
2011; Dalsgaard et al., 2012). The OMZ overlying the Peruvian shelf is a temporally
and spatially dynamic system where biogeochemical conditions can range from fully
oxic to anoxic and sulfidic. Occasional shelf oxygenation events occur mostly during
El Niño events and are linked to the propagation of coastal trapped waves (Gutiérrez
et al., 2008), but could be also observed during a coastal El Niño event (Lüdke et al.,
2019). During such events oxygen can reach down the upper slope to 200 m – 300 m
water depth (Levin et al., 2002). By contrast, sulfidic events can occur during periods





of stagnation, when not only oxygen but also $NO_3^-$ and $NO_2^-$ become depleted in the
water column due to low water mass exchange. Once $NO_3^-$ and $NO_2^-$ are depleted,
chemolithoautotrophic $H_2S$ oxidation is impeded. Under such conditions, $H_2S$
produced by bacterial sulfate reduction in sediments can be released into the water
column (Schunck et al., 2013), with the amount of $H_2S$ being released on the Peruvian
shelf reaching several mmol $m^{-2}$ $d^{-1}$ (Sommer et al., 2016)


**2. Methods**

**2.1 Sampling and sample handling**
In this study, data from three different types of samples were combined: (1) pore
waters for the determination of benthic diffusive fluxes and to study TM cycling in
sediments; (2) Benthic chamber incubations, to determine in-situ fluxes across the
sediment-water interface; (3) Near bottom water profiles to determine the fate of TMs
in the particle-rich and reactive benthic boundary layer.
Our sampling took place during RV Meteor cruises M136 and M137 in austral
autumn between April and May 2017. We also compare our recent data set to benthic
diffusive Fe(II) flux data from RV Meteor cruise M92, which took place in austral
summer during January 2013. Our sampling stations cover the entire Peruvian shelf
and slope across a transect at 12°S (Fig. 1). This transect covered water depths from
75 m to 950 m and includes stations above, inside and below the permanent OMZ. Our
sampling of pore waters and sample collection from benthic chamber incubations
generally followed the methodology described in Noffke et al. (2012).
Short sediment cores of 30 cm to 40 cm length were retrieved with a multiple
corer. Upon recovery, the cores were directly transferred into the ship's cool room
(4°C). The supernatant bottom water was instantly sampled and filtered through 0.2
μm cellulose acetate filters (Sartorius) and acidified to pH < 1 with subboiled distilled
$HNO_3$. The sediment cores were subsequently sampled in vertical sections in a glove
bag under Ar atmosphere to prevent any contact with oxygen. The sediment samples
were centrifuged to separate the pore waters from the sediment solid phase. Pore





waters were then filtered in another Ar-filled glove bag through 0.2 µm cellulose acetate
filters (Sartorius). An aliquot of 8 ml was acidified to pH < 1 with subboiled distilled
$HNO_3$ and stored in acid cleaned low-density polyethylene (LDPE) bottles for TM
analysis. Another aliquot was taken for analysis of $H_2S$ concentrations. Additional
sediment subsamples were collected in pre-weight cups for water content and porosity
determination as well as for Cd and organic C concentrations measurements in the
solid phase.

210        Benthic landers that consist of titanium frames and contain two circular benthic

chambers to conduct in-situ incubations were deployed on the seafloor (see Sommer
et al. (2009) for details). After placement of the lander on the seafloor, the benthic
chambers (internal diameter of 28.8 cm) were partially driven into the sediment,
covering a sediment area of 651.4 $cm^2$. The seawater volume that was enclosed in the
chamber varied between 12 l and 18 l, depending on the insertion depth of the chamber
into the sediment. Prior to incubation, the seawater contained in the chamber was
repeatedly replaced with ambient seawater to replace solutes and flush out particles,
which were mobilized during the insertion of the chamber into the sediment. Over the
incubation time of around 32 hours, 8 consecutive samples were filtered in-situ through
0.2 µm cellulose acetate filters (Sartorius) via peristaltic pumps and collected in quartz
glass tubes. All sampling tubes were acid cleaned prior to use to guarantee a TM clean
sampling. After recovery of the lander, the quartz glass tubes were transferred to the
laboratory and the obtained sample amount of 12 ml for each incubation time was
stored in acid cleaned LDPE bottles and acidified to pH < 2 with subboiled distilled
$HNO_3$. Other samples were collected simultaneously for analysis of nitrogen species.
The incubated sediments within the benthic chamber were sampled after recovery of
the lander and pore waters were extracted to analyze $H_2S$ concentrations for
comparison with pore water profiles from parallel MUCs.

229        To determine TM concentrations across the benthic boundary layer, we used

the lander to collect water samples at a distance of 0.5, 1.0, 2.0, 3.0 and 4.0 m from
the seafloor. Filter holders with 0.2 µm polyether sulfone filters (Supor) were attached
at the various depths and connected to sampling tubes that went through peristaltic
pumps into gas sampling bags (Tedlar). The sampling at distances of 3.0 m and 4.0 m
above the seafloor was realized by attaching the filter holders and tubing to an arm
that was automatically unfolded upon placement of the lander at the seafloor. The





peristaltic pumps transferred the seawater from their sampling depth into the sampling
bags over the same time period of the lander incubations of around 32 hours. This
resulted in an average sample volume of 1.5 l per depth. All filters, tubing and sampling
bags were acid cleaned prior to deployment to guarantee a TM clean sampling. Directly
after sample retrieval an aliquot of 60 ml was stored in acid cleaned LDPE bottles and
acidified to pH < 2 for TM analysis. Another aliquot was taken for analysis of silicic acid
$(Si(OH)_4)$.

**2.2 Analytical methods**
Concentrations of Fe(II) in pore waters were measured on board directly after
sample retrieval by photometry using the ferrozine method (Stookey, 1970). Other
geochemical parameters in our different samples were also determined photometrical
(U-2001 Hitachi spectrometer) using standard techniques (Grasshoff et al., 1999).
Hydrogen sulfide concentrations were determined using the methylene blue method
and silicic acid concentrations were determined using a heptamolybdate solution as
reagent. Concentrations of nitrogen species were determined by an auto-analyzer
(QuAAtro, SEAL Analytical) using sulfanilamide as reagent (Hydes et al., 2010).
For TM analysis of bottom water samples we followed the procedure described
by Rapp et al. (2017), where the TMs are pre-concentrated by a fully automated device
(SeaFAST). After raising the sample pH to 6.4 with an ammonium acetate buffer (1.5
M), a sample amount of 15 ml was loaded onto a chelating resin column, where the
seawater matrix was rinsed off, before the TMs were collected into 1ml elution acid (1
M subboiled $HNO_3$). Due to the smaller size of pore water samples and samples from
benthic lander incubations, a half-automated device (Preblab) with a smaller sample
loop and thus dead volume was applied for these sample types. On this device, sample
loading and collection as well as the addition of buffer was done manually. For samples
from benthic lander incubations, an amount of 3 ml and for pore waters 1 ml was
needed for pre-concentration. The samples were diluted with de-ionised water (MilliQ,
Millipore) to increase the sample volume to 5 ml for samples from benthic chamber
incubations and to 3 ml for pore waters. The pre-concentrated samples were measured
by ICP-MS (HR-ICP-MS; Thermo Fisher Element XR) and TM concentrations were
quantified by isotope dilution (Rapp et al., 2017). Accuracies for replicate
measurements of certified reference seawater for TMs are listed in Table 1.

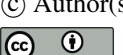



Cadmium and Al concentrations in sediments for calculation of sedimentary Cd
enrichments (Cd$_{xs}$) were determined following total digestions of freeze dried and
ground sediment. The sediment was digested in 40 % HF (suprapure), 65 % HNO3
(suprapure) and 60 % HClO$_4$ (suprapure). Concentrations were measured by ICP-OES
(VARIAN 720-ES). The reference standard MESS was used to check the digestion
procedure, the accuracy was ± 0.3 % for Cd and ± 1.3 % for Al (MESS-3 Cd: 0.24 ±
0.01 µg g$^{-1}$, recommended value 0.24 ± 0.01 µg g$^{-1}$, MESS-3 Al: 8.59 ± 0.11 µg g$^{-1}$,
recommended value 8.59 ± 0.23 µg g$^{-1}$).
Organic carbon content in the sediments was determined, after removal of
inorganic carbon with 0.25 mM HCl, using an Elemental Analyzer (Euro EA). Precision
of the measurement was ± 1 %.

**2.3 Flux calculations**
Benthic diffusive fluxes (F$_D$) were determined using Fick's first law of diffusion
using concentration gradients between the uppermost pore water sample (0 – 1 cm)
and the overlying bottom water (dC/dx) (Boudreau, 1997):
$$F_D = -\Phi D_{sed}(\mathrm{dC/dx}) \quad (1)$$

The effective molecular diffusion coefficients of Fe and Cd for sediments (D$_{sed}$) were
calculated from the molecular diffusion coefficient in seawater (D$_{sw}$) under standard
conditions (Li and Gregory, 1974) by adjusting it to in-situ temperature, pressure and
salinity applying the Stokes-Einstein Equation. We determined the diffusion
coefficients for sediments as follows:
$$D_{sed} = D_{sw}/\theta^2 \quad (2)$$

Tortuosity (θ) was calculated from porosity (Φ) as follows (Boudreau, 1997):
$$\theta^2 = 1 - \ln(\phi^2) \quad (3)$$

Positive values represent a flux from the bottom water into the sediment pore water.
The fluxes from benthic lander incubations were calculated from the slopes of
linear regressions, resulting from concentration changes over the incubation times.
Fluxes were corrected for water volume enclosed in the benthic chamber, which was



determined for each deployment from the insertion depth of the benthic chamber into
the sediment.


## 3. Results


### 3.1 Biogeochemical conditions in the water column

Cruise M136 and M137 took place in April and May 2017, during the decline of
a coastal El Niño event. A coastal El Niño is a local phenomenon that refers to reduced
upwelling and increased sea surface temperatures off the coasts of Peru and Ecuador,
with typically heavy rainfall on land. During this event in austral summer, coastal waters
off Peru showed a strong positive sea surface temperature anomaly of up to 2 to 4 °C
(Echevin et al., 2018; Garreaud, 2018). The warming is proposed to be a result of
strong local alongshore wind anomalies and equatorial Kelvin waves propagating
towards the Peruvian coast (Echevin et al., 2018; Peng et al., 2019). Due to the
particular atmospheric and oceanographic conditions, the water column overlying the
Peruvian shelf was oxygenated during our sampling campaign. Oxygen concentrations
were > 20 µM in the water column down to around 100 m water depth. However oxygen
concentrations in bottom waters directly above the seafloor on the shallowest station
(station 1) were below the detection limit (> 1 µM) measured through optopodes
attached to lander. The OMZ, with $O_2$ concentrations < 5 µM, extended from around
120 to 400 m water depth. The water column within the OMZ was nitrogenous (i.e.
$NO_3^-$ reducing) as indicated by the presence of $NO_2^-$ (≥ 4 µM), an intermediate product
of denitrification (Zumft, 1997). Oxygen gradually increased to > 50 µM from below 400
m towards 950 m water depth (Fig. 2). As we will compare some of our data to those
of an earlier cruise (M92), the corresponding oxygen distribution across the Peruvian
continental margin is shown for comparison (Fig. 2).

### 3.2 Benthic iron cycling

Iron concentrations in near bottom waters decreased from near shore to off
shore stations, from > 100 nM at the shallowest shelf station at 75 m water depth



(station 1) to 6 nM at 750 m water depth (station 9) (Fig. 3). At a number of stations
within the OMZ (stations 3 and 4), vertical concentration gradients were observed.
Here Fe concentrations decreased by 15 - 20 nM from 0.5 to 4 m above the seafloor.
Multiple sampling at the shallowest shelf station (station 1) revealed that Fe
concentrations were temporally variable and ranged from ~ 100 nM at the end of April
to < 60 nM at the end of May 2017.
Concentrations of Fe(II) in pore waters were highest (up to a few µM) in the
upper 5 – 10 cm of the sediment cores. Downcore, concentrations decreased to > 0.2
µM (Fig. 4). At all stations, concentrations in pore waters at the sediment surface were
higher than in the overlying bottom water, which implies a diffusive flux from pore
waters into bottom waters. The steepest concentration gradients across the sediment-
water interface were observed within the OMZ. The highest Fe(II) concentrations at
the sediment surface (> 6 µM) occurred at station 4 (145 m water depth). At this station,
the benthic diffusive flux into the bottom waters was also highest with 17.1 mmol m$^{-2}$
y$^{-1}$. The lowest diffusive fluxes of 0.0 (due to concentrations below the detection limit)
and 0.4 mmol m$^{-2}$ y$^{-1}$ were observed on the upper slope below the OMZ at stations 9
and 10 respectively (Table 2). An accumulation of $H_2S$ in pore waters coincided with a
depletion of Fe(II) concentrations (Fig. 4). At station 1, we observed the highest $H_2S$
concentrations throughout the core and in particular at the sediment surface, with
maximum concentrations reaching > 4 mM. At stations below the OMZ (stations 9 and
10), no $H_2S$ was detected within pore waters (Fig. 4).
Iron concentrations inside the benthic chambers were generally higher than in
ambient bottom waters, and reached maximum values > 300 nM. At stations 4 and 6,
located inside the OMZ, concentrations in the chambers increased in a linear way
during the incubation. At stations above and below the OMZ, we did not observe a
linearly increasing concentration trend over the incubation time. Following previous
studies (Turetta et al., 2005; Severmann et al., 2010; Noffke et al., 2012; Lenstra et al.,
2019) and for comparison with diffusive fluxes, we estimated benthic Fe fluxes from
linear regressions from the change in Fe concentrations in the benthic chamber against
incubation time (Table 2). To test the plausibility of the flux magnitude within the
chamber, we also calculated theoretical concentration gradients over time based on
our diffusive flux estimates (Fig. 5). Our incubation data were largely consistent in
direction and slope with the diffusive benthic fluxes. Especially at stations inside the



OMZ (station 4 and 6), where the highest diffusive fluxes of 17.1 and 8.3 mmol m$^{-2}$ y$^{-1}$
were observed, the projected and observed concentration gradients were in good
agreement. At stations below the OMZ, diffusive fluxes of < 1 mmol m$^{-2}$ y$^{-1}$ were too
low to be detected over our incubation time of 32 hours.


**3.3 Benthic cadmium cycling**
In near-bottom waters Cd concentrations increased with distance from the
coast, from 0.4 nM at the shallowest station at 75 m water depth (station 1) to 1.1 nM
below the OMZ at 750 m water depth (station 9). Cd concentrations were constant
between 0.5 and 4 m above the seafloor (Fig. 3).
Cadmium concentrations in pore waters ranged between 0.1 – 2 nM (Fig. 6).
Within the OMZ, bottom water concentrations were higher than concentrations in pore
water within the surface sediments (0 - 1 cm) indicating a downward diffusive flux into
the sediments. The benthic diffusive fluxes inside the OMZ were of the order of 0.5 –
0.8 µmol m$^{-2}$ y$^{-1}$ (Table 3). In contrast, at station1 and 9 an upward-directed
concentration gradient was observed, indicating a diffusive flux from the sediments into
bottom waters. The upward diffusive flux ranged from 1.9 above the permanent OMZ,
to 0.2 µmol m$^{-2}$ y$^{-1}$ below the OMZ (Table 3). Pore water Cd concentrations at greater
sediment depths were mostly higher than bottom water concentrations. In some cases
(station 3 and 4), elevated pore water Cd concentrations (up to 2 nM) coincided with
elevated H$_2$S concentrations (few hundred µM).
In the benthic chambers three different Cd patterns were observed (Fig. 7).
Above the permanent OMZ (station 1), Cd concentrations in the chambers were low (<
0.2 nM) throughout the incubation period, indicating that there was no Cd flux. At sites
within the OMZ (station 4, 5 and 6), concentrations decreased from ~ 0.6 nM to 0.3 nM
over the course of the incubation. Below the OMZ (stations 9 and 10), Cd concentration
in the chamber were high (~ 1 nM) and remained constant or increased slightly during
the incubation. At sites within the OMZ, Cd removal within the chamber was nearly
linear, which translates to a removal flux of 13 - 23 µmol m$^{-2}$ y$^{-1}$. The Cd removal fluxes
in benthic chambers were more than one order of magnitude higher than diffusive
benthic fluxes (0.5 – 0.7 µmol m$^{-2}$ y$^{-1}$) (Table 3).





## 4. Discussion


### 4.1 Benthic iron cycling

### 4.1.1 Comparison of diffusive and in-situ benthic chamber iron fluxes

In the absence of oxygen and, thus, bottom-dwelling macrofauna at stations within the OMZ, bioturbation and bioirrigation are unlikely to exert an important control on sedimentary Fe release. Consistent with this notion, the slope calculated from benthic diffusive fluxes is largely consistent with the concentration gradients observed within the benthic chambers (Fig. 5). Moreover, our fluxes from benthic chamber incubations and diffusive fluxes are generally of the same order of magnitude (few mmol m$^{-2}$ y$^{-1}$) (Table 2). Therefore, diffusive transport of dissolved Fe from the sediment into the bottom water seems to be the main control on the concentration evolution observed within the benthic chamber.

Some of the concentration gradients in benthic chambers are non-linear, indicating that the Fe flux was not constant during the incubations. This observation can be used to identify additional processes affecting Fe concentrations and fluxes within the benthic chamber, which may also affect fluxes under natural conditions. One possible process that can remove dissolved Fe(II) under anoxic conditions is Fe oxidation with $NO_3^-$ as the terminal electron acceptor or oxidation with $NO_2^-$ (Straub et al., 1996; Carlson et al., 2013; Klueglein and Kappler, 2013). The oxidation of reduced Fe in the absence of oxygen either biologically mediated by $NO_3^-$ or abiotically by $NO_2^-$ has been hypothesized to be important in the water column of the Peruvian OMZ (Scholz et al., 2016; Heller et al., 2017). During our incubation at station 4 (Fig. 8), we observed a decline in Fe concentrations during the first ten hours of the incubation time. Concurrently, $NO_3^-$ concentrations were decreasing, while $NO_2^-$ accumulated, presumably due to progressive denitrification and release from the sediments. Once $NO_3^-$ and $NO_2^-$ were quantitatively reduced, Fe concentrations started to rise again and the following concentration increase resulted in the highest in-situ Fe flux observed throughout our sampling campaign (Table 2). The coincidence in timing of Fe





accumulation and $NO_2^-$ decrease suggest that depletion of Fe at the beginning of the
incubation was most likely caused by Fe oxidation with $NO_2^-$. The incubation at station
4 was the only one where $NO_3^-$ and $NO_2^-$ were quantitatively removed during the
incubation. However, the high Fe flux cannot be interpreted as a natural flux estimate
at steady state. In general, we argue that bottom water $NO_2^-$ concentrations exert first
order control on the intensity of Fe efflux at the absence of oxygen and, therefore, need
to be considered in the evaluation of sedimentary Fe mobility in anoxic-nitrogenous
OMZs.
During the incubations at the stations 1, 9 and 10, Fe concentrations did not
continuously increase but varied between high and low values. This observation could
be explained by a combination of bioirrigation and bioturbation as well as rapid Fe
oxidation and precipitation. Under oxic conditions, bottom-dwelling macrofauna is likely
to increase the transfer of dissolved Fe from the sediments into the bottom water (Elrod
et al., 2004; Lenstra et al., 2019). During episodes of oxygenation a population of
macrofauna that can enhance bioturbation and bioirrigation was observed on the
Peruvian shelf (Gutiérrez et al., 2008). However, under oxic conditions, any Fe
delivered to the chamber is prone to rapid oxidative removal. Moreover, ex-situ
experiments have demonstrated a fast and efficient removal of up to 90% of dissolved
Fe in incubated bottom waters due to particle resuspension (Homoky et al., 2012).
Interactions with particles and oxidation processes can efficiently remove Fe shortly
after its transfer to bottom waters and this process is likely to be most intense close to
the seafloor where the highest particle concentrations prevail. We argue that the same
processes are reflected by declining Fe concentrations away from seafloor in some of
the bottom water profiles (at station 3 and 4) (Fig. 3).

**4.1.2 Removal rates of dissolved iron in the benthic boundary layer**
We observed declining Fe concentrations in the first 4 m away from the seafloor
at station 3 and 4, which hints at removal of dissolved Fe in the near bottom waters,
after its release from the sediments. To differentiate dilution with ambient bottom water
(by currents) from Fe removal from the dissolved phase, Fe concentrations were
normalized by silicic acid ($Si(OH)_4$) measured in the same samples (Fig. 3). Due to
opal dissolution within Peru margin sediments, silicic acid is released into bottom
waters (Ehlert et al., 2016). In contrast to Fe, we assume that silicic acid behaves



conservatively and precipitation reactions within the bottom waters are of subordinate
importance. The decreasing Fe to Si(OH)$_4$ ratios at station 3 and 4 with distance from
the seafloor indicate that there is Fe removal within the benthic boundary layer that
must be related to precipitation processes or scavenging.
We further constrained rates of dissolved Fe removal at stations with a
discernable Fe to Si(OH)$_4$ gradient within the first 4 m distance from the seafloor. To
this end, we first determined an eddy diffusion coefficient (K$_y$) using silicic acid fluxes
from benthic chamber incubations (F$_{Si}$) (chapter 2.3 for methodology) and the known
concentration gradient of dissolved silicic acid within the bottom water (d$_{Si}$/d$_x$), where
x is the height above the seafloor. At the seafloor, the flux of silicic acid from the
sediment is equal to the flux in the water column.
$$F_{Si} = -K_y(d_{si}/d_x) \quad (4)$$

This equation can be rearranged to find the eddy diffusion coefficient.
Dissolved Fe in the bottom water (DFe) can be described by the following
reaction-transport equation (ignoring advection and assuming a steady-state first-order
consumption of dissolved Fe):
$$DFe = C_{BW} * exp.(-\sqrt{KFeox}/\sqrt{Ky}) \quad (5)$$

The equation can be fitted to the measured DFe concentrations in the bottom water by
adjusting the Fe concentration directly above the seafloor (C$_{BW}$) and the Fe oxidation
constant (K$_{feox}$). From the fitted first-order rate constant K$_{Feox}$, we can determine the
half-lives for dissolved Fe in bottom waters.
The half-lives of dissolved Fe in the first 4 m away from the seafloor are 2.5 min
and 0.3 min at station 3 and 4, respectively (Table 4). Another study reported a
dissolved Fe half-life of 17 hours under nitrogenous conditions in the first 10 to 20 m
above the seafloor in the Peruvian OMZ (Scholz et al., 2016). Our calculations suggest
that Fe removal in near-bottom waters is much faster. As mentioned above, in the
absence of oxygen, removal processes could be related to oxidation of dissolved Fe
with NO$_2^-$ or to interactions with suspended particles, which are likely to be most
abundant directly above the seafloor. Further research on dissolved-particulate
interactions in bottom waters is needed to better constrain how sedimentary Fe fluxes
are modified in the benthic boundary layer.





### 4.1.3 Controls on the temporal variability of benthic iron fluxes

The Peruvian OMZ is known to experience high-amplitude fluctuations in upwelling intensity as well as bottom water oxygen, $NO_3^-$, $NO_2^-$ and $H_2S$ concentrations (Pennington et al., 2006; Gutiérrez et al., 2008; Graco et al., 2017; Ohde, 2018). To get an insight into how different biogeochemical conditions control benthic diffusive Fe(II) fluxes, we compare the fluxes from our recent cruise with fluxes from our earlier cruise M92 (Fig. 9). Cruise M92 took place in austral autumn 2013 following the main upwelling season and during a period of intense primary productivity. Due to reduced upwelling and stable density stratification, the water column on the shallow shelf was not only depleted in oxygen but also in $NO_3^-$ and $NO_2^-$ during cruise M92 (Sommer et al., 2016). Under such conditions, chemolithoautotrophic $H_2S$ oxidation with $NO_3^-$ or $NO_2^-$ was impeded so that pore water $H_2S$ could be released from the sediment into the water column. As a result, the water column during M92 was sulfidic between around 50 and 150 m water depth with the highest $H_2S$ concentration of 13 μM observed at 50 m depth (Fig. 2). While the biogeochemical conditions on the shallow shelf were fundamentally different to those during M136 and M137, below 150 m water depth the conditions were largely comparable (oxygen-depleted, $NO_3^-$: 20 – 30 μM, $NO_2^-$ up to 9 μM between 150 – 300 m). At the stations with similar biogeochemical water column conditions, the Fe(II) fluxes during both our sampling campaigns were remarkably similar (Fig. 9). However, similar to the temporal variability of Fe concentrations in bottom waters at station 1 (Fig. 3), we observed a pronounced difference in the diffusive flux magnitude on the shallow shelf, where the biogeochemical conditions differed between both cruises. The highest diffusive flux during M92 in 2013 of 22.7 mmol $m^{-2}$ $y^{-1}$ was measured at station 1. By contrast, during M136/137 in 2017 we observed a much lower flux of 2.6 mmol $m^{-2}$ $y^{-1}$ at this station. During M136 and M137 the highest flux of 17.1 mmol $m^{-2}$ $y^{-1}$ was measured at station 4, located at 145 m water depth.

Diffusive fluxes are a function of the concentration gradient between pore water and bottom water (Eq. (1)). As dissolved Fe concentrations in bottom waters are generally much lower compared to those observed in pore waters, the flux magnitude is chiefly determined by differences in pore water Fe concentrations. During M92, pore waters at the sediment surface were characterized by high dissolved Fe concentrations





(4.8 µM in the upper pore water), which resulted in a steep gradient and a comparably high Fe flux. Under the slightly sulfidic conditions that prevailed in the water column during M92, oxidative removal of dissolved Fe(II) with $NO_3^-$ or $NO_2^-$ was impeded (Scholz et al., 2016) and dissolved Fe(II) could be stabilized as aqueous iron sulfide (Schlosser et al., 2018). Therefore, the bottom water was characterized by high dissolved Fe concentrations (up to 0.7 µM in MUCs overlying bottom water).

Despite oxic conditions in the water column during M136 and M137, we observed much higher $H_2S$ concentrations in surface sediments at station 1 compared to M92 (4100 µM during M136 and M137 versus 1800 µM during M92 within the first 8 cm of the core) (Fig. 4). Because of higher $H_2S$ concentrations, Fe concentrations were controlled by the solubility of Fe monosulfide minerals (FeS). It may seem counterintuitive that the surface sediment was highly sulfidic, while the overlying water column was oxygenated. In order to explain this observation, we need to consider the role of mats of filamentous sulfur oxidizing bacteria in controlling $H_2S$ concentrations in surface sediments. (Gutiérrez et al., 2008; Noffke et al., 2012; Yücel et al., 2017). During M92 these mats were generally abundant on the shelf and upper slope, thus limiting the extent of $H_2S$ accumulation within surface sediments (Sommer et al., 2016). Previous studies demonstrated that mats of sulfur oxidizing bacteria can disappear during periods of oxygenation (Gutiérrez et al., 2008). Consistent with this previous finding, visual inspection of the seafloor using the video-guided MUC revealed that the abundance of bacterial mats on the seafloor seemed greatly reduced, which is most probably related to oxic bottom water conditions on the shallow shelf during the coastal El Niño event. As these microaerophilic organisms tend to avoid high oxygen concentrations they probably started to withdraw into the sediment once oxygen levels raised. Furthermore, an abundance of red squat lobster (*Pleuroncodes monodon*), which are known to feed on bacterial mats (Gallardo et al., 1994), was observed at the seafloor on the shallow shelf. We suggest that the retreat of sulfide-oxidizing bacteria under oxic conditions created a situation where $H_2S$ accumulation in the surface sediment and FeS precipitation limited the extent of Fe release into the bottom water.

**4.2 Benthic cadmium cycling**

**4.2.1 Comparison of diffusive and in-situ benthic chamber cadmium fluxes**



At stations above the permanent and below the OMZ (stations 1, 9 and 10),
diffusive Cd fluxes and fluxes in benthic chambers were largely consistent (Table 3).
In contrast, the fluxes determined with benthic chambers at stations within the OMZ
(station 4, 5 and 6) were 25 to 40 times higher than the diffusive flux (Table 3). This
discrepancy demonstrates that diffusion cannot be the dominant process leading to
the continuous decrease of dissolved Cd during benthic chamber incubations.
Alternatively, Cd could be precipitated within the benthic chamber and removed
through downward sinking of Cd-rich particles. Cadmium sulfide (greenockite) has a
relatively low solubility compared to sulfide minerals of other TMs (solubility product of
CdS = -16.4 << FeS = -3.9). It is generally agreed that CdS precipitation can takes
place at trace amounts of $H_2S$ ($H_2S$ < 1 µM, i.e., below the detection limit of the method
applied in this study) (Davies-Colley et al., 1985; Rosenthal et al., 1995). Previous
studies using in-situ benthic flux chambers have concluded that production of $H_2S$ in
the sediment or the accumulation of $H_2S$ in benthic chambers during incubations can
switch the direction of the Cd flux or intensify Cd removal through CdS precipitation
(Westerlund et al., 1986; Colbert et al., 2001). Precipitation of CdS during the
incubation is, therefore, a viable explanation for the discrepancy between diffusive Cd
flux and Cd fluxes in benthic chambers observed in our study. Furthermore, the three
different pattern of Cd concentration trends observed in benthic chamber incubations
can be related to $H_2S$ concentrations in the surface sediment below the benthic
chambers (Table 3). At stations within the OMZ, pore water $H_2S$ concentrations in
surface sediments were moderate (few µM). It is likely that there was a continuous
leakage of trace amounts if H2S from the pore water into the incubated bottom waters
during the incubation thus leading to CdS precipitation and declining Cd
concentrations. On the shallowest shelf station, where pore water $H_2S$ concentrations
in the surface sediment were high (hundreds of µM), a potentially large amount could
have been released at the beginning of the incubation thus explaining pronounced Cd
depletion in the chamber compared to the surrounding bottom water (0.1 nM within the
chamber compared to 0.4 nM outside the chamber). Below the OMZ, where there was
no $H_2S$ present in surface sediments, there was no Cd depletion in the chamber during
the incubation and, consistent with previous studies in oxic settings (Westerlund et al.,
1986; Ciceri et al., 1992; Zago et al., 2000; Turetta et al., 2005), both diffusive and
benthic chamber flux data were indicative of an upward-directed flux out of the
sediment. Due to the absence of $H_2S$, dissolved Cd released from biogenic particles in



the surface sediment could accumulate in the pore water thus driving a diffusive flux
out of the sediment.

**4.2.2 Quantification of the sedimentary cadmium sink**
Consistent with our Cd flux data there is general consent that OMZs are a sink
for Cd. Several water column studies have observed Cd depletion in water masses
within the Peruvian and other OMZs, which was mostly attributed to Cd removal via
CdS precipitation in sulfidic micro-niches within particles in the water column (Janssen
et al., 2014; Conway and John, 2015b; Xie et al., 2019). Sedimentary studies showed
that Cd is highly enriched in OMZ sediments, which has mostly been attributed to the
delivery of Cd with organic material and subsequent fixation as CdS within sulfidic
sediments (Ragueneau et al., 2000; Böning et al., 2004; Borchers et al., 2005; Muñoz
et al., 2012; Little et al., 2015). Based on our data, we can quantify the delivery of Cd
to the sediments via three different pathways: (1) diffusion across the sediment-water
interface and CdS precipitation within the sediment; (2) Cd incorporation by
phytoplankton and delivery to the sediment with organic matter; (3) CdS precipitation
in the water column and particulate delivery to the sediment (Table 3).
The enrichment of Cd in the sediment relative to the lithogenic background
(expressed as excess Cd concentration, $Cd_{xs}$) was calculated using the following
equation (Brumsack, 2006):

$$Cd_{xs} = Cd_{sample} - Al_{sample} * (Cd/Al)_{crust} \quad (6)$$

The Cd/Al ratio of the upper continental crust ($1.22*10^{-6}$) was used as lithogenic
background reference (Taylor and McLennan, 2009). To get a flux of Cd to the
sediment, $Cd_{xs}$ was multiplied with the mass accumulation rate (MAR) from published
data for each individual site (Dale et al., 2015b). To approximate the amount of Cd
delivered to the sediment with organic material, the average concentration ratio of Cd
to C in phytoplankton (Moore et al., 2013) was multiplied by published particulate
organic carbon rain rates (maximum estimate) or burial rates (minimum estimate) for
each individual site (Dale et al., 2015b). The Cd delivery via precipitation in the water
column was determined as the remainder of $Cd_{xs} * MAR$ after subtraction of the two
other sources (i.e., diffusive flux and delivery by organic material).

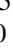



Sediments at all stations on the Peruvian shelf and slope are enriched in Cd
relative to the lithogenic background. The accumulation rate of $Cd_{xs}$ decreases from
250 µmol m$^{-2}$ y$^{-1}$ at the station 1 at 75 m to 4 µmol m$^{-2}$ y$^{-1}$ at station 9 at 750 m water
depth (Table 3). These fluxes generally exceed the amount of Cd delivered to the
sediments via diffusion and associated with organic material. Together these
mechanisms of Cd delivery can only account for ~ 20 % of the $Cd_{xs}$ at stations above
and inside the OMZ, with the delivery with organic material being of greater importance.
The remaining water column Cd removal (~80 %) must be related to CdS precipitation
in the water column and delivery of Cd-rich particles to the sediment. This removal
process can be a combination of precipitation in sulfidic micro-niches around sinking
particles (Janssen et al., 2014; Bianchi et al., 2018), CdS precipitation in sulfide plumes
(Xie et al., 2019) when sedimentary H$_2$S can spread throughout the water column
(Schunck et al., 2013; Ohde, 2018), and precipitation of CdS in the near-bottom water
(this study). Our estimated CdS precipitation in the water column within the OMZ agree
with the Cd fluxes we determined from benthic chamber incubations, where dissolved
Cd removal takes place in the first 20 cm – 30 cm away from the sediment surface.
These Cd removal fluxes alone are sufficient to account for 41% – 68% of the
estimated particulate Cd removal from the water column and 38% - 60% of total $Cd_{xs}$
in the sediment within the OMZ (Table 3). Considering that Cd precipitation in near-
bottom water is unlikely to be restricted to the 20 – 30 cm above the seafloor covered
by our benthic chambers, the removal flux associated with this process is likely to be
even higher. Below the OMZ at 750 m, where the smallest Cd enrichment is observed,
the relative contribution of Cd delivery with organic material increases. About half of
the $Cd_{xs}$ can derive from organic material at this station.
Once Cd is delivered to the sediment it can either stay fixed in the solid phase
or it can be released into the pore waters. Cadmium concentrations in pore waters of
subsurface sediments (> 10 cm sediment depth) were mostly higher than bottom water
concentrations (Fig. 6), indicating a transfer of Cd from the solid phase into pore waters
during early diagenesis. Cadmium sulfides are considered highly insoluble and stable
within sediments (Elderfield et al., 1981), even upon re-oxygenation (Rosenthal et al.,
1995). Therefore, Cd release through re-dissolution of CdS is ruled out as a potential
source of dissolved Cd. Alternatively, Cd liberation upon remineralization of organic
material could explain elevated Cd concentrations in the pore water. Elevated Cd
concentrations in sulfidic pore waters have been observed in previous studies and



attributed to Cd stabilization through formation of organic and inorganic complexes
(Gobeil et al., 1987; Sundby et al., 2004). Experimental data gave evidence for the
presence of dissolved Cd bisulfide and polysulfide complexes in pore waters. An
increase of electrochemically active Cd after UV irradiation, was explained by the
destruction of electrochemically inactive bisulfide and polysulfide complexes (Gobeil et
al., 1987). At very high $H_2S$ concentrations ($> 10^{-3}$ M) the solubility of Cd may increase
due to an increase in these bisulfide and polysulfide complexes. Under such highly
sulfidic conditions, Cd solubility may even exceed the solubility in oxygenated waters
and highly sulfidic sediment can eventually turn into a diffusive source of Cd to the
bottom water (Davies-Colley et al., 1985). Such a scenario can explain the negative
(i.e., upward-directed) diffusive Cd flux at station 1, where the pore waters of surface
sediments are highly sulfidic.


**5. Conclusions and implications for trace metal sources and sinks in the future ocean**

Consistent with earlier work, our results demonstrate that that OMZ sediments
are a source for Fe and a sink for Cd. Moreover, our findings allow to further constrain
the different biogeochemical conditions and processes that control the benthic fluxes
of these TM across the Peruvian OMZ.
Iron is transported via diffusion from the sediment pore water into bottom water.
The accumulation of high levels of $H_2S$ in pore waters, modulated by the abundance
of sulfur oxidizing bacteria, can reduce diffusive Fe release through sulfide precipitation
within pore waters. In anoxic bottom waters Fe can be rapidly removed, likely via
oxidation by $NO_2^-$ and/or interaction with particles. Benthic Cd fluxes are directed from
the bottom water into the sediment within the OMZ. Diffusive fluxes and delivery of Cd
via organic material cannot account for the sedimentary Cd enrichment. Instead CdS
precipitation in near bottom waters could be the most important pathway that delivers
Cd to the sediments.
According to our results, $H_2S$ concentrations in surface sediments exert a first
order control on the magnitude and direction of Fe and Cd fluxes across the sediment-
water interface. With generally decreasing oxygen concentrations in the ocean and an



expansion of OMZs (Stramma et al., 2008; Schmidtko et al., 2017), sulfidic surface
sediments will likely also expand. With regard to the solubility of their sulfide minerals,
Fe and Cd represent two opposite end members. The solubility of sulfide minerals of
other important nutrient-type TMs, such as Ni and Zn, is intermediate between those
of Fe and Cd (Fe > Ni > Zn > Cd). An expansion of sulfidic surface sediments is thus
likely to affect sedimentary TM fluxes in a differing manner. This notion is illustrated in
Fig. 10, showing saturation indices calculated based on the range of TM
concentrations observed in the ocean and typical $H_2S$ concentrations observed in
anoxic marine environments (nM - μM concentrations represent sulfidic events in the
water column, pore water conditions are represented by up μM - mM concentrations).
Cadmium sulfide minerals become oversaturated at nM to μM $H_2S$ concentrations,
which is why Cd removal can take place in the bottom water in OMZs. By contrast, FeS
is highly undersaturated under the typical biogeochemical conditions in the water
column. Therefore, FeS precipitation is unlikely to take place in the water column, even
under somewhat more reducing conditions. Other sulfide-forming TMs have an
intermediate sulfide solubility, which could imply that the direction and magnitude of
their sedimentary fluxes is susceptible to expanding ocean anoxia. The differing
response of TMs to an expansion of sulfidic conditions may cause a change in the TM
stoichiometry of upwelling water masses with potential consequences for TM-
dependent marine ecosystems in surface waters.


**Data availability**
The data will be made available at Pangaea upon publication of the article.


**Author contribution**
AP and FS conceived the study. AP, FS, AD, SS conducted the sampling at sea. AP
analyzed the trace metal concentrations. AP and FS prepared the manuscript with
contributions from all co-authors.




**Competing Interests**

The authors declare that they have no conflict of interest.


**Acknowledgements**

We are grateful for the support of the crew of RV Meteor during the fieldwork. For their technical and analytical assistance we thank A. Beck, A. Bleyer, B. Domeyer, A. Petersen, T. Steffens, R. Surberg and M. Türck. This study was supported by the German Research Foundation through the Emmy Noether Nachwuchsforschergruppe ICONOX (Iron Cycling in Continental Margin Sediments and the Nutrient and Oxygen Balance of the Ocean) and Sonderforschungsbereich 754 (Climate-Biogeochemistry Interactions in the Tropical Ocean).



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





**Figure captions**

Fig. 1: Bathymetrical map of the Peruvian continental margin and sampling stations along the latitudinal depth transect at 12° S. The sampling stations for pore water profiles are depicted by white stars, for bottom water profiles by yellow dots and for benthic chamber incubations by red dots.

Fig. 2: Oxygen, nitrate, nitrite and hydrogen sulfide profiles on the Peruvian slope (1000 m depth), crossing the oxygen minimum zone (upper panel), and the upper shelf (75 m depth) (lower panel) during cruises M136 & M137 and M92 along the 12° S transect.

Fig. 3: Near bottom water profiles of dissolved Fe and Cd concentrations and dissolved Fe to silicic acid ratios in the benthic boundary layer 0.5 m to 4 m above the seafloor across the 12° S transect. Depicted by the red diamond is a second sampling with a time difference of one month at station 1.

Fig. 4: Pore water profiles of dissolved Fe(II) and hydrogen sulfide concentrations. For station 7 (300 m water depth) and station 10 (950 m water depth) pore water profiles are not shown to save space, but the diffusive fluxes are listed in table 2. The profile of an earlier cruise, M92, at station 1 (75 m water depth) is displayed for comparison. The uppermost sample of each profile represents the bottom water concentration. All symbols are within error.

Fig. 5: Dissolved Fe concentrations in incubated bottom waters from benthic chamber incubations. The dashed line represents theoretical concentration gradients over the incubation time based on our benthic diffusive fluxes (Table 2). All symbols are within error.

Fig. 6: Pore water profiles of dissolved Cd concentrations. The uppermost sample of each profile represents the bottom water concentrations. All symbols are within error.

Fig. 7: Dissolved Cd concentrations in incubated bottom waters from benthic chamber incubations. The dashed line represents theoretical concentration gradients over the incubation time based on our benthic diffusive fluxes (Table 3). All symbols are within error.



Fig. 8: Dissolved Fe, nitrate and nitrite concentrations in incubated bottom waters
from the benthic chamber incubation at station 4 (145 m water depth).
Fig. 9: Comparison of benthic diffusive Fe(II) fluxes between cruises M136 & M137
and M92 on the Peruvian shelf. Negative values represent fluxes from the sediment
pore water into the bottom waters. Shaded bars on the upper panel display the
geochemical conditions in the water column during the time of sampling.
Fig. 10: Schematic overview of how the mobility of different trace metal may respond
to an expansion of sulfidic conditions. Saturation indices (SI) were calculated for
different $H_2S$ concentrations and reported minimum and maximum concentrations of
trace metals in the water column (data from Bruland and Lohan 2003). Solubility
products for Fe (FeS ppt), Ni (Millerite), Zn (Wurtzite), Zn (Greenokite) were taken
from the Pitzer database (Plummer et al., 1988).The results are approximate since
concentrations instead of activities were used for calculations. A positive SI is
indicative of oversaturation whereas a negative SI is indicative of undersaturation.











1139                                    Figure 2





Table 1: Accuracy values for replicate concentration measurements (n = 7) of certified
reference seawater for trace metals NASS-7 and CASS-6 by ICP-MS.

|  | NASS-7 certified value | NASS-7 measured value | CASS-6 certified value | CASS-6 measured value |
|---|---|---|---|---|
| Fe (µg/L) | 0.351 ± 0.026 | 0.352 ± 0.017 | 1.56 ± 0.12 | 1.56 ± 0.03 |
| Cd (µg/L) | 0.0161 ± 0.0016 | 0.0162 ± 0,0024 | 0.0217 ± 0.0018 | 0.0216 ± 0.0016 |




















Table 2: Comparison of benthic diffusive Fe(II) fluxes out of the sediment and
geochemical bottom water conditions between M92 and M136 & M137 on the
Peruvian shelf. Fluxes during M92 correspond to similar depth (see Fig. 9).

| station | M136 & M137 | M136 & M137 | M136 & M137 | M136 & M137 | M136 & M137 | M136 & M137 | M92 | M92 |
|---|---|---|---|---|---|---|---|---|
| | water depth | latitude | longitude | bottom water condition | Fe(II) flux diffusive | Fe flux benthic chamber | bottom water condition | Fe(II) flux diffusive |
| | (m) | (S) | (W) | | ($mmol\ m^{-2}\ y^{-1}$) | ($mmol\ m^{-2}\ y^{-1}$) | | ($mmol\ m^{-2}\ y^{-1}$) |
| 1 | 75 | 12°13.52 | 77°10.93 | $O_2 < 5\ \mu M$ | -2.56 | -1.74 | slightly sulfidic | -22.69 |
| 3 | 130 | 12°16.68 | 77°14.95 | nitrogenous | -0.81 | - | slightly sulfidic | -3.16 |
| 4 | 145 | 12°18.71 | 77°17.80 | nitrogenous | -17.08 | -8,57 | nitrogenous | -5.77 |
| 5 | 195 | 12°21.50 | 77°21.70 | nitrogenous | -2.72 | 2.01 | nitrogenous | -1.51 |
| 6 | 245 | 12°23.30 | 77°24.82 | nitrogenous | -8.31 | -5,43 | nitrogenous | -10.20 |
| 7 | 300 | - | - | nitrogenous | -3.02 | - | nitrogenous | -3.12 |
| 9 | 750 | 12°31.35 | 77°35.01 | $O_2 > 5\ \mu M$ | 0.00 | -6.11 | $O_2 > 5\ \mu M$ | 0.4 |
| 9 | 970 | 12°34.90 | 77°40.32 | $O_2 > 5\ \mu M$ | -0.25 | -1.68 | $O_2 > 5\ \mu M$ | -0.12 |














Table 3: Comparison of sedimentary Cd excess compared to the lithogenic
background and the contribution of Cd delivery to the sediment via different
pathways: (1) diffusion across the sediment-water interface and Cd sulfide
precipitation within the sediment; (2) Cd incorporation by phytoplankton and delivery
to the sediment with organic matter; (3) Cd sulfide precipitation in the water column
and particulate delivery to the sediment.

| station | water depth | Cd excess sediment[1] | Cd flux diffusive | Cd flux benthic chamber | $H_2S$ surface in sediment below benthic chamber | Cd from organic matter[2] | CdS precipitation in water column[3] |
|---|---|---|---|---|---|---|---|
| | (m) | ($\mu$mol m$^{-2}$ y$^{-1}$) | ($\mu$mol m$^{-2}$ y$^{-1}$) | ($\mu$mol m$^{-2}$ y$^{-1}$) | ($\mu$M) | ($\mu$mol m$^{-2}$ y$^{-1}$) | ($\mu$mol m$^{-2}$ y$^{-1}$) |
| 1 | 75 | 248.87 | -1.85 | -1.6 (3109.5)* | 641.02 | 8.34 - 49.04 | 199.83 – 240.53 |
| 3 | 130 | 153.41 | 0.83 | - | - | 4.87 – 17.40 | 135.19 – 147.72 |
| 4 | 145 | 35.07 | 0.52 | 13.4 | 1.30 | 1.55 – 6.48 | 28.07 – 32.99 |
| 5 | 195 | 44.76 | 0.72 | 22.6 | 9.52 | 5.71 – 7.71 | 36.36 – 38.36 |
| 6 | 245 | 35.15 | 0.55 | 21.2 | 0.40 | 3.60 – 6.54 | 28.06 – 31.00 |
| 9 | 750 | 4.44 | -0.24? | 0 | 0 | 1.48 – 3.21 | 1.23 – 2.96 |


[1] Calculated after Brumsack (2006) and multiplied with the mass accumulation rate
for each site (Dale et al., 2015b).
[2] Determined by multiplication of Cd to C ratio in average phytoplankton (Moore et al.,
2013). For maximum values organic carbon rain rates and for minimum values
organic carbon accumulation rates (Dale et al., 2015b) were used.
[3] Remainder of Cd excess in sediment after subtraction of diffusive and organic Cd
sources.
* Flux calculated from gradient of Cd bottom water concentration (0.5 m) and
concentration in first benthic chamber incubation sample (0.25 h).




Table 4: Modelled half-lives ($t_{1/2}$) of dissolved Fe within the first 4 m distance from the
seafloor at station 3 (130 m water depth) and station 4 (145 m water depth) and data
used for determination of $t_{1/2}$ using Eq. (4) and Eq. (5).

| station | water depth | Si(OH)$_4$ flux benthic chamber ($F_{Si}$) | Si(OH)$_4$ concentration gradient ($d_{Si}$) | Eddy diffusion coefficient ($K_y$) | Modelled Fe at sediment surface ($C_{BW}$) | Fe oxidation constant ($K_{Feox}$) | Half-live in benthic boundary layer ($t_{1/2}$) |
|---|---|---|---|---|---|---|---|
| | (m) | ($\mu$mol cm$^{-2}$ d$^{-1}$) | ($\mu$mol cm$^{-3}$ cm$^{-1}$) | (m$^2$ s$^{-1}$) | (nM) | (d$^{-1}$) | (min) |
| 3 | 130 | 0.73 | $-4.05{*}10^{-6}$ | $1.55 10^{-6}$ | 70 | 400 | 2.5 |
| 4 | 145 | 0.33 | $-1.44 10^{-6}$ | $1.96 10^{-6}$ | 81 | 3500 | 0.3 |




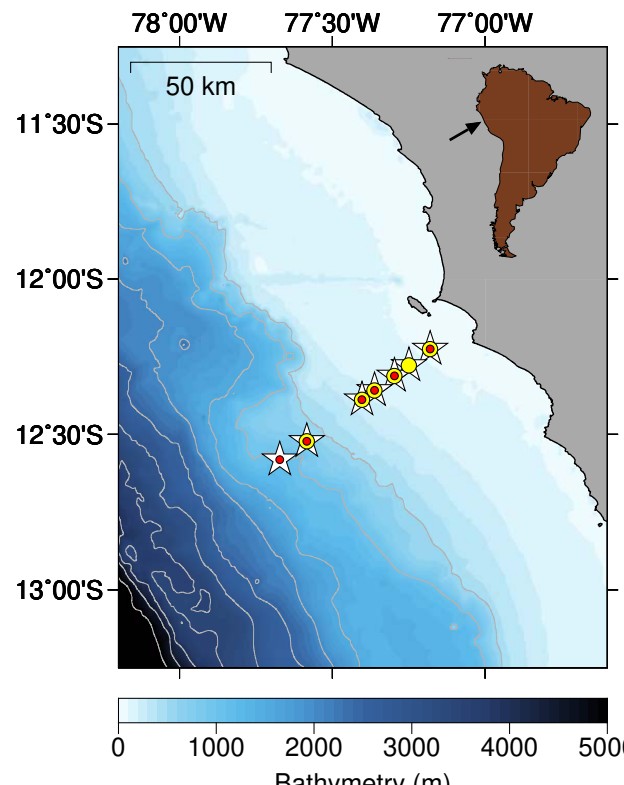

Figure 1



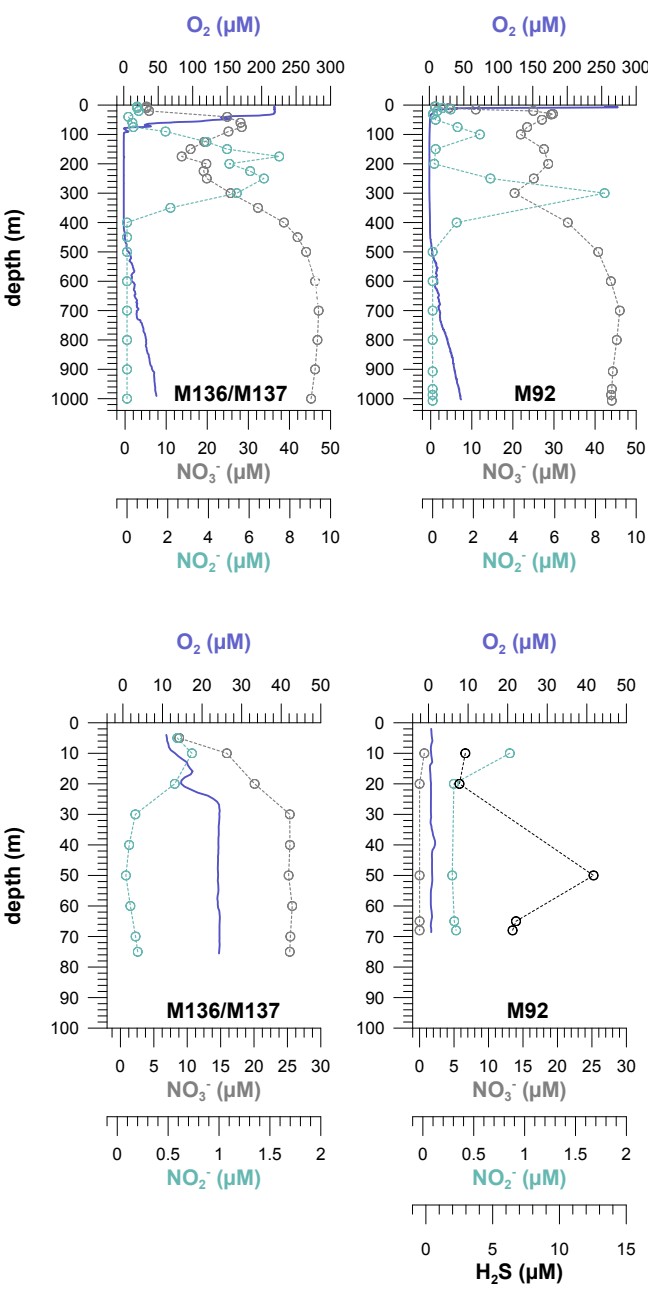

Figure 2





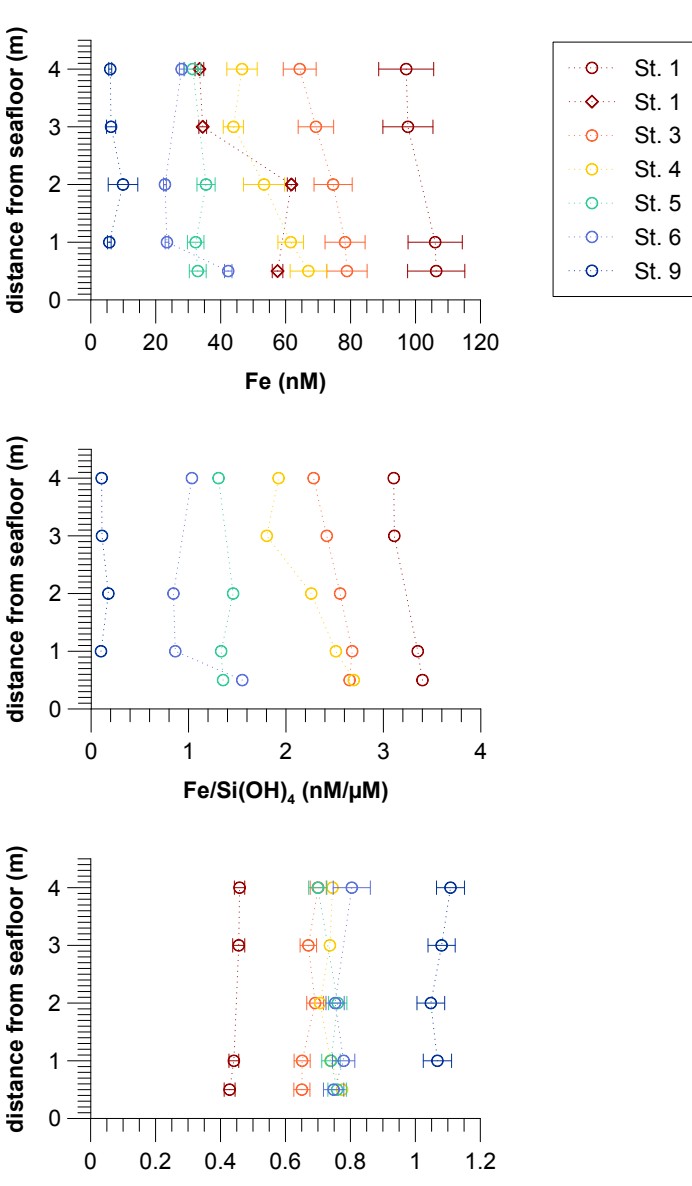

Figure 3



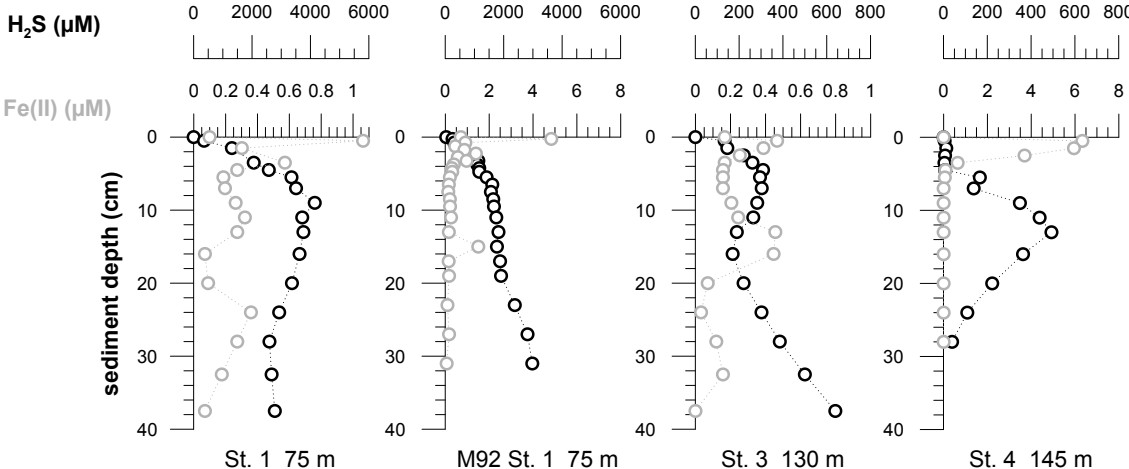

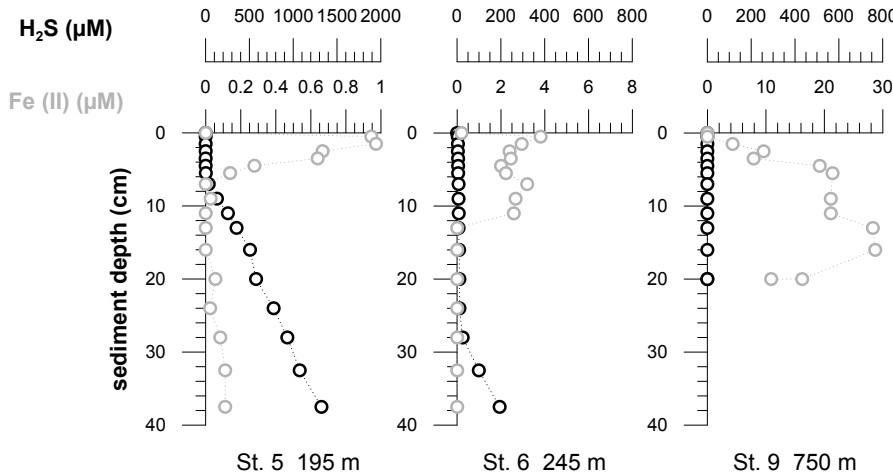

Figure 4




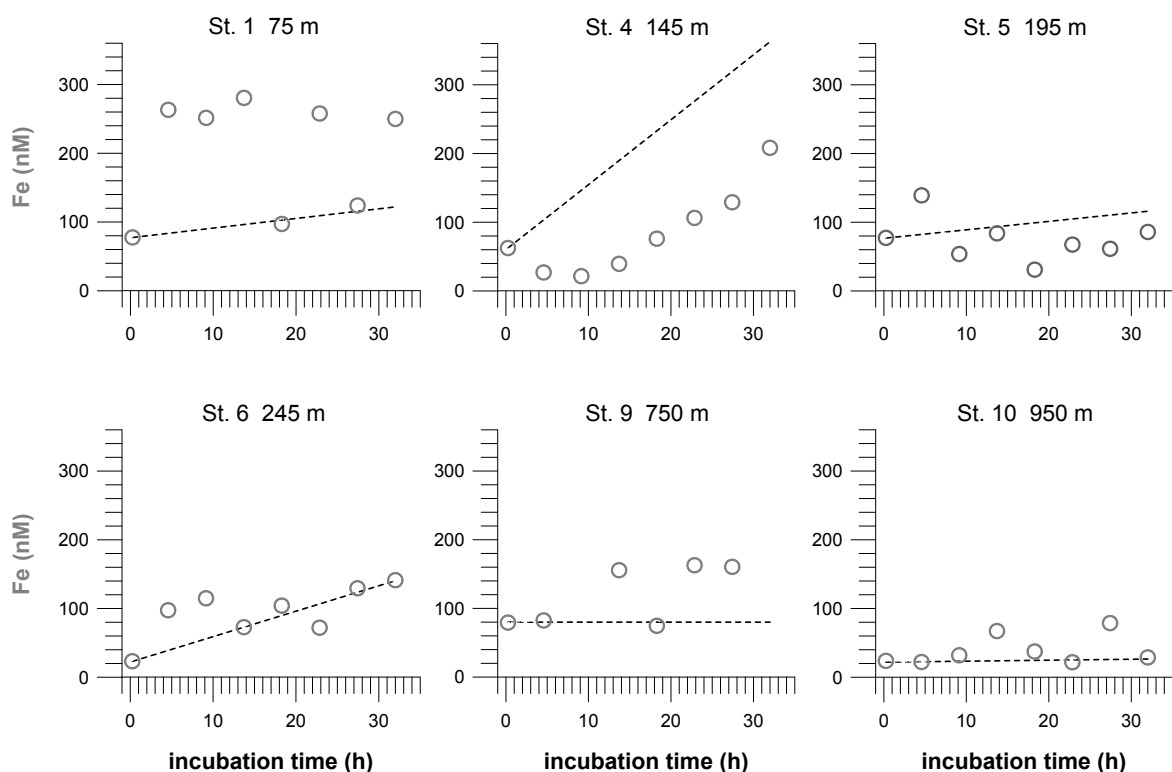

Figure 5





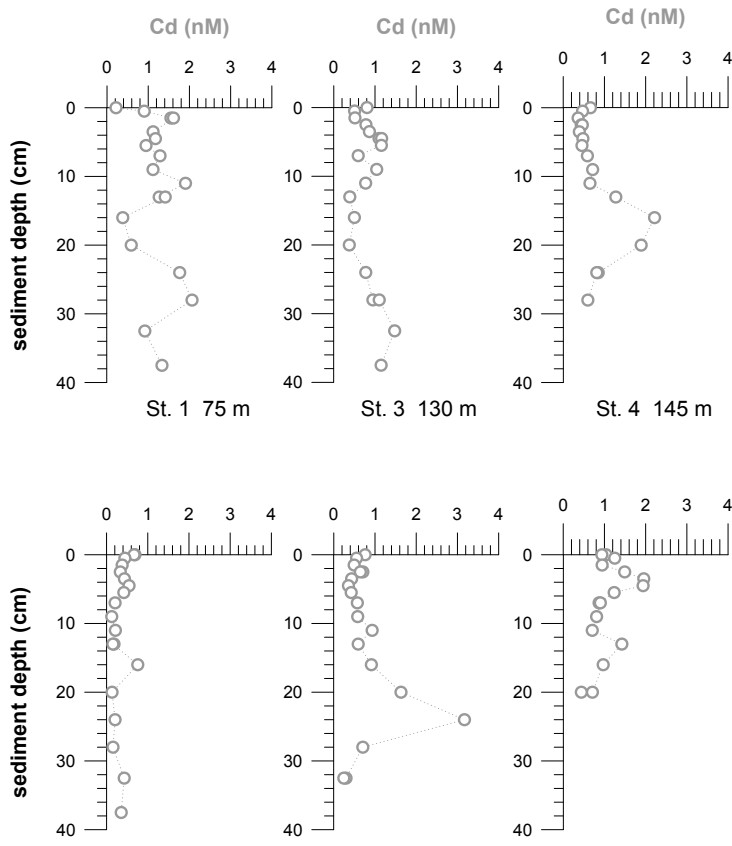

Figure 6



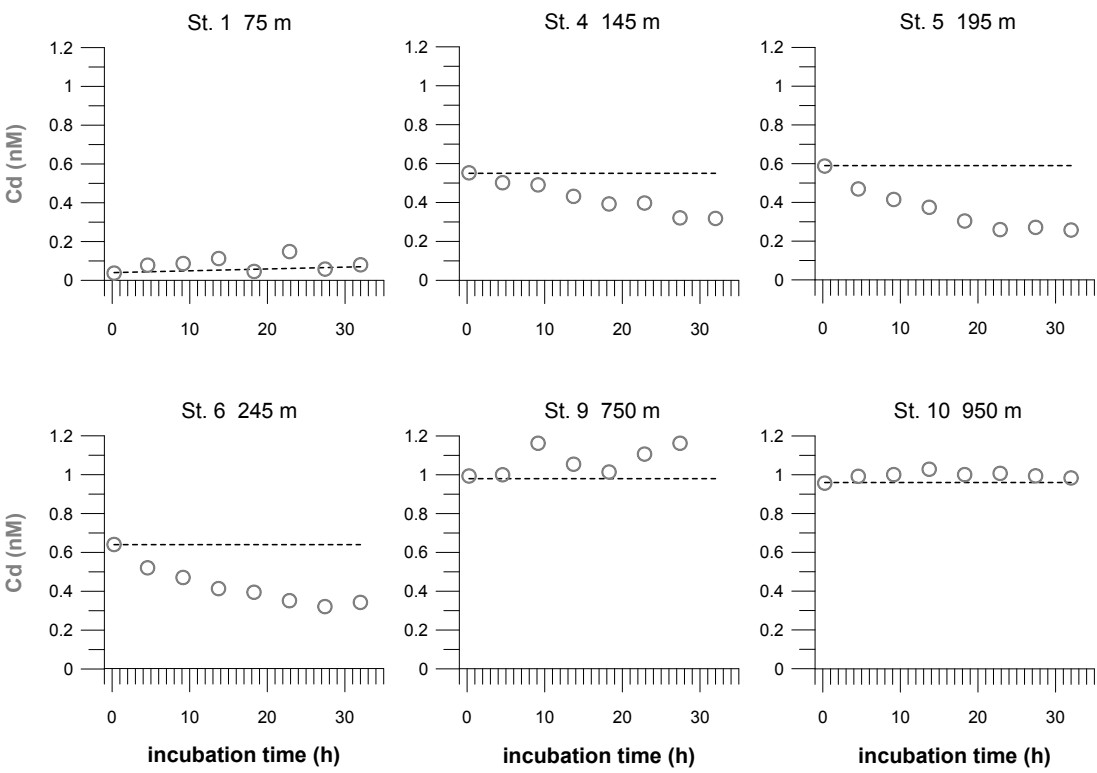

Figure 7



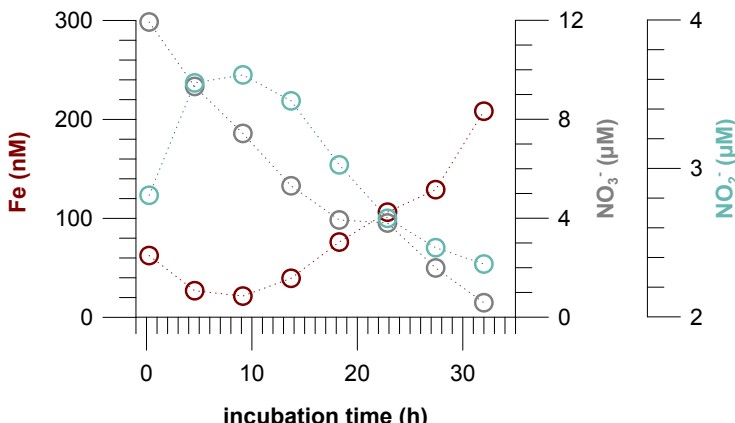

Figure 8



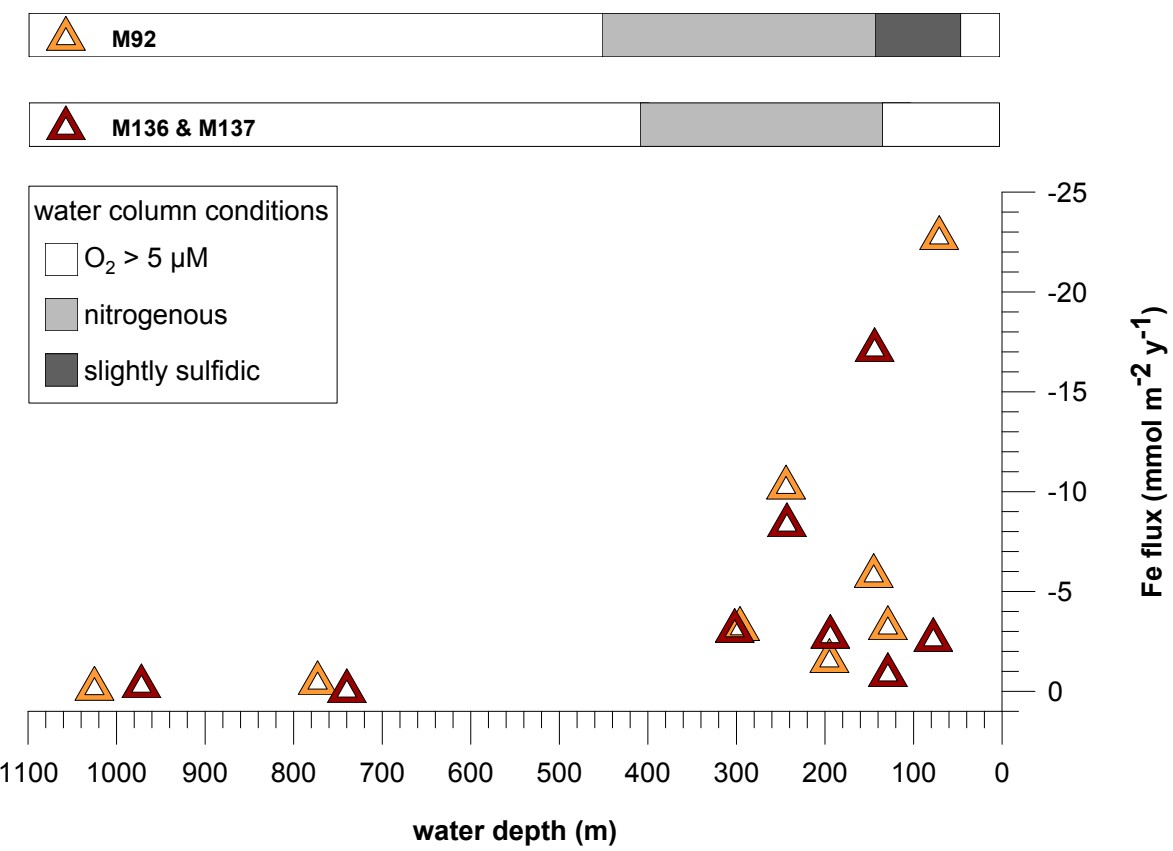

Figure 9



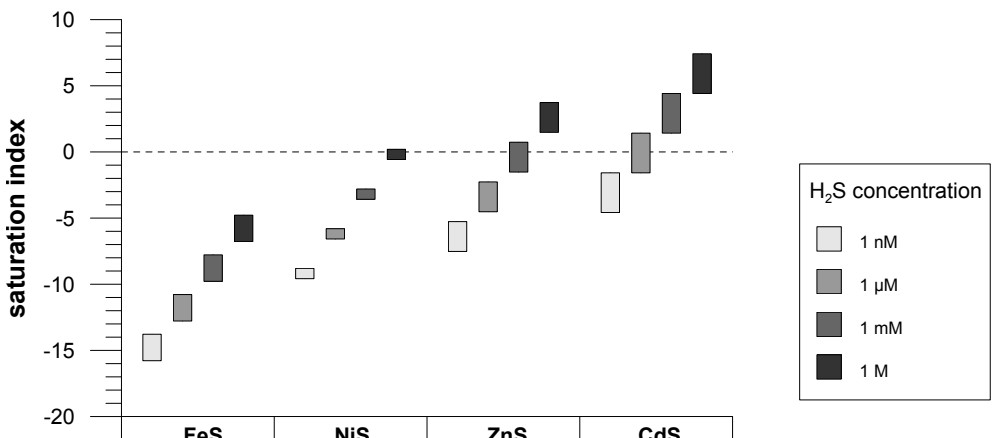

Figure 10