# Peer review of "The control of hydrogen sulfide on benthic iron and"

_Biogeosciences, 2019_

## Referee Comment (RC1) · Edouard Metzger (Referee) · 1 Dec 2019

Dear editor,

This is my contribution to help Biogeosciences and the authors of this manuscript to improve the diffusion of this very valuable work. I did the best according to my knowledge and understanding to be fair in my criticism and hope tha this piece of work will be published. However I think that there is still some work to be done.

Sincerely

Dr. Edouard Metzger

[Figure]

General comments The main concern of this study is the precision of flux modeling from chemical profiles as for porewaters as for near-bottom waters. Since mas balance calculations and main conclusions are made out of these data, I would suggest a more extended presentation of methods, results to allow reader to better evaluate the quality of discussion and conclusions made.

Some reorganization could be made in the introduction to show better how what is known about metals and OMZ functioning and what can be added from the study. It would also help to better specify hypotheses made and features predicted on benthic fluxes of trace elements.

The importance of TM benthic fluxes is stated in the abstract and the conclusion as important for surface marine ecology. Without suggesting any intention from the authors, I believe that from the present study to such statement the shot is really too long and this may mislead the less careful reader towards rapid conclusions appealing a bit to its emotion instead of its reason

Specific comments

Abstract L41-42 I am not sure to follow the causality between metal relative metal solubility and spatial-temporal heterogeneity L45-47 the last sentence of the abstract tends to suggest that decrease of cadmium solubility due to sulfide increase consequently to oxygen declining may affect marine ecosystems. The sentence appears to me overstated, out of the scope of what your data can say, overall, as a general sentence that would appeal to emotion. I would suggest to find another sentence to take perspective. Maybe adding another metal that you must have analyzed usinc a ICPMS to the study showing different sulfide affinities would have helped to strengthen such a statement.

Introduction All along the introduction, it is stated that little is known about TM benthic fluxes in hypoxic environments. I would disagree with that. The authors cite themselves several of them and I think of studies done by Faganeli and coworkers in the Adriatics, work done during the Microbent project in Thau lagoon including by myself. Work in

hypoxic estuaries such as the Gironde or the Scheldt among others. I have the feeling that if authors had detailed more about TM behavior in different environments during occurrence of hypoxia/anoxia and sulfide release into the water column they could have made a series of predictive hypotheses on what would be Cd behavior in the Peruvian OMZ. Overall, I am not a fan of the outline of the introduction because the rationale is at the beginning and the state of the art shown after is too far to make the reader quite sur about what are really the hypotheses made here and what is really new.

Methods Incubation time was of 32h, how oxygen evolved within chambers that were not anoxic in the beginning?

How the benthic boundary layer's thickness is established and what is its dynamics? That should be mentioned in the introduction as how it may affect benthic fluxes What was the detection and quantification limit for Cd measurements

I am aware that there is plenty of literature with diffusive flux calculations from overlying water and the first porewater concentrations. I think this is wrong as soon as a 2-point calculation is subject to the precision of those points and overlying water concentration is an average of 10 to 30 cm of water column according to the fullness of the core. If any author took the entire porewater profile to make an averaged concentration for gradient calculation nobody would take the calculation seriously. I would at least be skeptical about calculations taking only two points to model a line from which, one is an average of something it is impossible to fully describe with such sampling method. I would say that this can be overtaken adding a supplementary point for the gradient determination. Unfortunately, only one of your profiles can apply. This aspect must be clearly discussed in the manuscript and conclusions made from such calculations carefully done. High resolution methods exist.

Linear regressions with standard deviation of the slope are not shown in the document, this should appear in the graphics with also error bars for each point

Particulate flux calculations and CdS water column uptake calculations should also be

detailed here, especially the last one.

Results

I think that section 3.1. appeals a lot to literature to be part of the result section. Only bottom oxygen data is described from figure 2. I woud suggest authors to discard the figure 2, table 2 is sufficient, and put that description in the methods sections For a result section I would avoid naming sections using processes such a "biogeochemical cycling". I would suggest to simply call it: "Porewater and benthic fluxes" then "iron, then "cadmium"

I would also suggest to avoid citations. Why citations are provided for iron and not for cadmium? Some symmetry should be maintained between these elements at least here to underline differences in results

In the methods section I made a comment about the thickness of the benthic boundary layer. According to the 4 metres profile from the bottom, it appears that the benthic boundary layer is clearly thinner than 0.5 m as concentrations are stable in your date for almost all profiles. I believe from in situ data of oxygen profiles that this boundary layer is within the range of few micrometers to few centimeters. I am not sure those data are really relevant here and could be discarded or remain as supplementary data. Line 351 authors say that initial incubation concentrations were higher than "bottom samples". This pleads to the fact that those bottom samples did not describe the benthic boundary layer. This should be said somehow Did the authors thought of showing modelled gradients in the porewater profiles instead of showing modelled incubation points from diffusive fluxes?

Discussion

4.1.1. Talking about incubations 1, 9 and 10 you mention bioturbation as a potential artefact, what about oxygen consumption and sulfide precipitation during incubation? Actually a combination of all that could happen as oxygen depletion could bring endobionts to surface (see riedel et al experiments in the adriatics) or enhance bioirrigation (Duport et al, 2007) favoring sulfide efflux or iron precipitation. Did you consider leaks in the chamber or dysfunctional homogenization? This is a current issue with in situ incubation. Oxygen and sulphide data from incubations are really missing here to state about ventilation processes or leaking

4.1.2. I am not convinced about the half-life calculations since there is little change in iron concentrations and they do not fit with a reaction transport model to me. What about these data being colloidal fractions that pass the 0.2 $\mu$m poresize of the filter that would have little reactivity as shown by the slight slopes that look mostly within error bars? I agree that 1 point from station 8 and two from station 1 are clearly above error bar range but only station 8 seems to indicate that you caught the BBL. However, station 1 profile points to 2 homogeneous layers that suggest diffusion is not dominant there Silica data should appear somewhere

4.2.1. Line 632 is it possible to detail how CdS precipitation in the near bottom water column was calculated? Show linear regressions, statistical significance and determination coefficients. What are the ranges in table 3? This section is very interesting but I feel uncomfortable with the lack of details about calculations and numbers out of it. It is a pity because it is I guess the post important part of the study

Conclusions Station 1 seems to weaken the first statement of the conclusion The order of TM affinity towards sulfide is shown for the frst time in the conclusion. It should be said more explicitly in the introduction

The expansion of OMZ is evocated to draw a future scenario of TM burial/recycling within the OMZ but what about seasonality and other cyclic controls such as el nino/la nina conditions? Conclusions seem a bit too bold in that way. I am not sur that a putative enhancement or decrease of TM upward flux to surface waters (that is by the way not shown here with the Cd example) would affect marine ecology. At least nothing in the paper discusses it. If I read only the conclusion and the abstract, this will be the

take-home message of the paper I would guess and the paper does not address such important question.

Technical corrections

Line 189 "compared" Line 191 "covered" Line 207 "weighted" Line 317 what are optopodes? Line 563 "can take"

Figures 5 and 7 should show linear regressions and equations as well as error bars for the data points. Then, equations and quality of regressions should be quickly described to prepare the discussion section about iron being oxidized by nitrogen compounds

Details from saturation indices shown in the caption of figure 10 should maybe appear within the text of the manuscript in the methods or the discussion

Table 3 what is within brackets for station 1 flux from chambers. This calculation would show what evolution of Cd concentration within the chamber? Is that coherent with your data?

Figure 9 reminds me a figure from my paper about cadmium fluxes in a seasonal hypoxic lagoon (Metzger et al., 2007) with a kind of threshold. You could put in the x-axis oxygen concentration and or sulfide and or nitrate. Why did you do only a figure for iron?

---

## Referee Comment (RC2) · Michael Staubwasser (Referee) · 3 Feb 2020

I generally agree with most of the authors conclusions, particularly with respect to the Cd data, but I do have a few issues concerning the calculation of diffusive Fe (and Cd) fluxes and the discussion of the Fe data. To adress these, I suggest the authors expand and adjust their discussion to deal with these aspects.

1) The input data required to verify the authors' flux calculations from concentration gradients between the uppermost pore water sample and the water column sample is absent from the manuscript's figures and tables. That includes the porosity used in equation 3 and the tortuosity calculated therefrom in eq. 2. Nor are the input concen-

tration values given. I cannot glean them from fig. 4, as the scale is too small and most pore water profiles appear to show an enormous spread in the uppermost two data points. What values were used for the calculation? Which one is the true surface in fig. 4? What is the reason for this spread in pore water concentrations so close to the surface - e.g. St1_M136 (Fe: $\sim$ 100 nM and >1 nM) and St4_M136 (Fe: low nM range? and > 6 $\mu$M). Comparing bottom water column data (fig. 3) does not suggest one of these plotted pore water values could actually be the bottom water concentration. I think it would be better, to calculate concentration gradients from more than just two data points for Fe and Cd. Perhaps the authors would also consider to show the 0-500 nM range in high resolution, and the few numbers above at lower resolution with a break in the axis between?

2) I also don't like the way these gradient-based diffusive flux calculations (lines) are shown over the flux chamber time series data in fig. 5. Better use the water column data as a starting point - since chambers were flushed with ambient bottom water - than the first chamber time series value. There often is quite a difference between the ambient bottom water and the first chamber value that the authors do not explain. Sometimes that difference is hard to understand. In Sta 1 ambient water is higher than the starting value of the chamber (flushed with ambient water). At Sta 9, chamber values ($\sim$ 80 nM) are much higher throughout than ambient water ($\sim$5 nM), but apparently there is no diffusive flux of Fe. That discrepancy does not agree with the authors' general statement in the abstract that the two flux estimates agree with each other.

3) Generally, I see an unaddressed problem in that the Fe concentration data for the near-bottom water column and the flux chambers were filtered at 0.2 $\mu$m and acidified for later analysis. That would dissolve all colloidal Fe, which I would expect to be quite abundant near the bottom (see e.g. Fitzsimmons et al. 2015, Mar Chem 173 pp 125 for a similar problem in ocean water underneath dust plumes, and Heller et al. 2017, GCA 211, pp 174 for a discussion of particulate - and colloidal - removal of dFe in the Peruvian OMZ). I am aware of the authors citing the latter manuscript, but they

do not cite it in the context of colloidal/particulate removal of dFe. I realize, that the necessary ultra-filtration to quantify colloidal Fe was not implemented, and of course, cannot be provided in hindsight. However, I would like to ask the authors to deal with this problem in the discussion. There is so much scatter in the Fe concentration time series from the flux chambers, that I am certain the dFe data suffer from colloids. Colloidal Fe would not count for flux calculations based on pore water - bottom water concentration gradients, and would very much complicate flux estimates based on the linear trend calculation from the flux chamber time series. My quick suggestion would be to provide two estimated for fluxes based on minimum and maximum estimates, but perhaps the authors can come up with a better way to deal with this issue. A discussion of this aspect would really benefit the manuscript. I suggest to discuss this aspect at the end of paragraph 4.1.1. In addition, bioirrigation should contribute considerably to resuspension of particles and colloids as well - a process which the authors briefly mention in lines 441-446 in general terms but do not seem to connect to their scattered flux chamber data.

4) In the abstract, lines 20-23, I would suggest the authors phrase the suggested "agreement" of different flux calculations somewhat more careful. This is not a very convincing statement even without considering the colloid problem. At best, there is agreement for some of the stations.

5) In the conclusion, I don't think it is appropriate to claim that Fe just diffuses out of the sediment column. The authors themselves have made the case for biorrigation and bioturbation as a significant process for Fe enrichment in the flux chambers.

Technical aspects that should be corrected: a) Please show cruise AND station number in figures 1 and 2. b) I believe it would be useful to show the H2S data along with the Cd data in fig. 6, since they are discussed together.

---

## Author Comment (AC1) · 17 Feb 2020

Dear Edouard Metzger, we would like to thank you a lot for reviewing our manuscript and your constructive comments.

In this response letter *referee comments are listed in italic font*, followed by the authors responses to each individual comment. Changes that will be made to the manuscript are underlined.

Referee comment 1:

*Abstract L41-42 I am not sure to follow the causality between metal relative metal solubility and spatial-temporal heterogeneity L45-47 the last sentence of the abstract tends to suggest that decrease of cadmium solubility due to sulfide increase consequently to oxygen declining may affect marine ecosystems. The sentence appears to me overstated, out of the scope of what your data can say, overall, as a general sentence that would appeal to emotion. I would suggest to find another sentence to take perspective. Maybe adding another metal that you must have analyzed usinc a ICPMS to the study showing different sulfide affinities would have helped to strengthen such a statement.*

Response to referee comment 1:

We are using the elements Fe and Cd as role models, as they have a contrasting sulfide solubility, and interpolate their behavior to trace metals which have an intermediate sulfide solubility (lines 86 – 92, 686 – 689, 699 - 701). The concentration/reservoir of a trace metal in the ocean is dependent on the respective input and output fluxes of this element. Consequently, benthic recycling and burial processes are an important aspect of the oceanic mass balance. As especially Fe and also other trace metals can often be the limiting factor for primary productivity, the availability of trace metals has a direct impact on phytoplankton growth. Where bottom water is upwelled to the surface ocean, the benthic burial or release of a trace metal can limit or fuel primary productivity. We are currently working on a second paper where we will present aspects on sedimentary cycling and benthic fluxes of other trace metals off Peru including Zn, which has a low sulfide solubility and Co which has a high sulfide solubility, that are in agreement with the conclusions of the present manuscript.

Referee comment 2:
*Introduction All along the introduction, it is stated that little is known about TM benthic fluxes in hypoxic environments. I would disagree with that. The authors cite themselves several of them and I think of studies done by Faganeli and coworkers in the Adriatics, work done during the Microbent project in Thau lagoon including by myself. Work in hypoxic estuaries such as the Gironde or the Scheldt among others. I have the feeling that if authors had detailed more about TM behavior in different environments during occurrence of hypoxia/anoxia and sulfide release into the water column they could have made a series of predictive hypotheses on what would be Cd behavior in the Peruvian OMZ. Overall, I am not a fan of the outline of the introduction because the*

*rationale is at the beginning and the state of the art shown after is too far to make the reader quite sur about what are really the hypotheses made here and what is really new.*

Response to comment 2:
It was not our purpose to disregard earlier work on sedimentary trace metal dynamics in inner-shore and coastal environments like lagoons and estuaries. In sections 1.2 and 1.3 we refer to earlier studies on benthic trace metal biogeochemistry and we will include further citations to studies in inner-shore environments such as  Metzger et al., 2007 and Point et al., 2007. We note, however, that except for Fe, little work has been done in open marine systems where the redox- and sediment-dynamics are different compared to estuaries and lagoons. The expansion of our knowledge on trace metal cycling and fluxes in these open-marine environments is the goal of our study.

In the general section on trace metal behavior (1.1 Scientific rationale) we will refer more explicitly to the knowledge gained in studies on estuaries and lagoons that trace metals can precipitate as sulfides when redox conditions change and $H_2S$ concentrations increase.

Referee comment 3:
*Methods Incubation time was of 32h, how oxygen evolved within chambers that were not anoxic in the beginning?*

Response to referee comment 3:
The evolution of oxygen concentrations during the benthic lander incubations will be published by Clemens et al. (in preparation).

The incubated bottom water in chambers below the oxygen minimum zone (station 9 and 10) did not become anoxic. At station 9 (750 m) the oxygen concentrations decreased from around 3 µmol at the beginning of the incubation to around 2 µmol at the end of the incubation. At the station 10 (950 m) oxygen remained above 10 µmol throughout the incubation time. For the incubation at station 1 (75 m) we cannot exclude that traces of oxygen were present during the first 5 hours of the incubation (some air bubbles may have been trapped inside the chamber during the lowering through the water column, that then dissolved during the beginning of the incubation). This may have reduced dissolved Fe concentrations in the first sample of this incubation through oxidative removal, however excluding this first value would not affect the overall flux magnitude derived from this incubation.

Referee comment 4:
*How the benthic boundary layer's thickness is established and what is its dynamics? That should be mentioned in the introduction as how it may affect benthic fluxes What was the detection and quantification limit for Cd measurements*

Response to referee comment 4:
It was our goal to sample and to capture the cycling of trace metal cycling in the highly dynamic layer overlying the seafloor. The common definition of the benthic boundary

layer is "a discrete layer of flowing sea water above a benthic substrate, delimited vertically by its contact with free stream flow" (D.J. Wildish, in Encyclopedia of Ocean Sciences (Second Edition), 2001). We acknowledge that, strictly speaking, our sampling interval does not correspond to the benthic boundary layer in the proper sense. We therefore decided to refer to "bottom-near water column" in the revised version.

For Cd measurements we applied the method by Rapp et al., 2017, where the detection limit for Cd is 0.8 pmol $L^{-1}$, which is well below the lowest concentrations of our samples (40 pmol L-1). The detection limit of Cd will be listed in section 2.2 Analytical methods.

Referee comment 5:
*I am aware that there is plenty of literature with diffusive flux calculations from overlying water and the first porewater concentrations. I think this is wrong as soon as a 2-point calculation is subject to the precision of those points and overlying water concentration is an average of 10 to 30 cm of water column according to the fullness of the core. If any author took the entire porewater profile to make an averaged concentration for gradient calculation nobody would take the calculation seriously. I would at least be skeptical about calculations taking only two points to model a line from which, one is an average of something it is impossible to fully describe with such sampling method. I would say that this can be overtaken adding a supplementary point for the gradient determination. Unfortunately, only one of your profiles can apply. This aspect must be clearly discussed in the manuscript and conclusions made from such calculations carefully done. High resolution methods exist.*

Response to referee comment 5:
For the diffusive flux calculations we deliberately decided to use the two point concentration gradient between the uppermost pore water sample and the overlying bottom water. We know that this method has its limitations, but need to stay with this approach for the following reasons.

1. This is a commonly used approach which allows us to compare our diffusive fluxes to fluxes from other publications (e.g. Sundby et al., 1986; Warnken et al., 2001; Turetta et al., 2005; Pakhomova et al., 2007; Noffke et al., 2012; Lenstra et al., 2019).

2. The strong gradients between the uppermost pore water and bottom water sample is typically observed within open-marine oxygen minimum zones (e.g. Noffke et al., 2012; Scholz et al., 2019). Since $H_2S$ accumulates close to the surface in these settings, the dissolved Fe peak in pore water is very narrow (order of 1 - 2 cm) and located close to the sediment-water interface. More advanced curve fitting methods would fail to capture sharp gradients at the sediment surface and would, therefore, yield erroneous flux estimates (see also Shibamoto and Harada, 2010; Dale et al., in prep.)

We will explicitly include the above explanation in section 2.3 Flux calculations.

Referee comment 6:
*Linear regressions with standard deviation of the slope are not shown in the document, this should appear in the graphics with also error bars for each point*

Response to referee comment 6:
We will plot the linear regressions for the concentration change over time during the benthic chamber incubation for Fe and Cd (figures 5 and 7). We will include the coefficient of determination for each linear regression. In figures where error bars are not shown, all symbols are within error as stated in the figure captions.

Referee comment 7:
*Particulate flux calculations and CdS water column uptake calculations should also be detailed here, especially the last one*

Response to referee comment 7:
The calculations used for the quantification of the sedimentary Cd sink and the different delivery pathways to the sediment are explained in section 4.2.2 Quantification of the sedimentary Cd sink. How we determine the particulate delivery of Cd to the sediment via CdS precipitation in the near-bottom water column is explained below table 3 (Cd enrichment in the sediment after subtraction of diffusive and organic Cd sources). We will also mention this more explicitly in the main text of section 4.2.2 Quantification of the sedimentary Cd sink.

Referee comment 8:
*I think that section 3.1. appeals a lot to literature to be part of the result section. Only bottom oxygen data is described from figure 2. I woud suggest authors to discard the figure 2, table 2 is sufficient, and put that description in the methods sections For a result section I would avoid naming sections using processes such a "biogeochemical cycling". I would suggest to simply call it: "Porewater and benthic fluxes" then "iron, then "cadmium"*

Response to referee comment 8:
In section 3.1 we present our measured data of oxygen, nitrate and nitrite during both cruises. The prevailing biogeochemical conditions are important for the trace metal cycling- That is why it is crucial to leave this data in the manuscript. Only regarding the very special oceanic condition, the occurrence of a coastal El Niño during our cruise in 2017, we refer to literature instead of our own data as this includes atmospheric and oceanographic observations that we could not conduct ourselves.

We fully agree to change the titles of the results sections as you suggested in 3.2 Bottom water, pore water and benthic flux data and then in subsections 3.2.1 Iron and 3.2.2 Cadmium.

Referee comment 9:
*I would also suggest to avoid citations. Why citations are provided for iron and not for*

*cadmium? Some symmetry should be maintained between these elements at least here to underline differences in results*

Response to referee comment 9:
We will remove the first part of the sentence starting in line 354 including the citations in line 355 -356.

Referee comment 10:
*In the methods section I made a comment about the thickness of the benthic boundary layer. According to the 4 metres profile from the bottom, it appears that the benthic boundary layer is clearly thinner than 0.5 m as concentrations are stable in your date for almost all profiles. I believe from in situ data of oxygen profiles that this boundary layer is within the range of few micrometers to few centimeters. I am not sure those data are really relevant here and could be discarded or remain as supplementary data. Line 351 authors say that initial incubation concentrations were higher than "bottom samples". This pleads to the fact that those bottom samples did not describe the benthic boundary layer. This should be said somehow Did the authors thought of showing modelled gradients in the porewater profiles instead of showing modelled incubation points from diffusive fluxes?*

Response to referee comment 10:
As mentioned in response to referee comment 4 we will replace the term 'benthic boundary layer' by the 'bottom-near water column'.

Regarding our statement in line 351 (comparing Fe concentrations in the bottom waters collected inside and outside of the benthic chambers): Different Fe concentrations inside and outside of the chamber are to be expected because of diffusive Fe release from the sediment and an accumulation in the enclosed water volume inside the benthic chamber. Furthermore, the bottom water enclosed in the benthic chamber is much closer to the seafloor compared to the bottom water samples collected closest to the seafloor (at 0.5 m). We assume that the concentrations in the bottom water enclosed in the chamber represents the average concentration over the first 30 cm above the seafloor. Outside of the chamber, a concentration gradient between the seafloor and the bottom-near water column is likely to establish because of diffusive release of Fe from the sediment and removal in the bottom water. Therefore, it is not surprising that different concentrations are observed in the chamber and in the bottom water 50 cm above the seafloor.

Furthermore, the water sample at 0.5 m away from the seafloor was collected in sampling bags with peristaltic pumps over a time span of 32 hours. This means that, in contrast to the benthic chamber, where samples were taken within minutes at discrete points in time, short-term concentration changes cannot be seen.

The above explanations will be added to the discussion in section 4.1.1 Comparison of diffusive and in-situ benthic chamber iron fluxes.

As mentioned in response to referee comment 6 we will plot linear regressions for the concentration change over time during the benthic chamber incubation for Fe and Cd

(figures 5 and 7) together with the slopes calculated from pore water profiles to facilitate the comparison of benthic fluxes derived by these two different approaches.

However, applying a transport-reaction model, which would need to involve carbon, sulfur, nitrogen and iron turnover to generate accurate pore water profiles of dissolved Fe, is beyond the scope of the present paper.

Referee comment 11:
*4.1.1. Talking about incubations 1, 9 and 10 you mention bioturbation as a potential artefact, what about oxygen consumption and sulfide precipitation during incubation? Actually a combination of all that could happen as oxygen depletion could bring endo-bionts to surface (see riedel et al experiments in the adriatics) or enhance bioirrigation (Duport et al, 2007) favoring sulfide efflux or iron precipitation. Did you consider leaks in the chamber or dysfunctional homogenization? This is a current issue with in situ incubation. Oxygen and sulphide data from incubations are really missing here to state about ventilation processes or leaking*

Response to referee comment 11:
Hydrogen sulfide could not be detected within bottom waters from benthic chamber incubations with our method, which has a detection limit of 1 µmol L$^{-1}$. For Fe to precipitate as sulfide several hundred µmol L$^{-1}$ of H$_2$S are required. Therefore, we exclude Fe sulfide precipitation within the benthic chambers.

As mentioned in our response to referee comment 3, we cannot exclude a slight oxygen contamination within the benthic chamber at station 1 during the very beginning of the incubation (some air bubbles may have been trapped inside the chamber during the lowering trough the water column, that then dissolved during the beginning of the incubation). This may have reduced dissolved Fe concentrations in the first sample of this incubation through oxidative removal. However, excluding this first value would not affect the overall flux magnitude derived from this incubation.

At stations below the permanent oxygen minimum zone (station 9 and 10) the oxygen was not consumed during the incubations, which is why it is unlikely that endobionts came to the sediment surface.

Referee comment 12:
*4.1.2. I am not convinced about the half-life calculations since there is little change in iron concentrations and they do not fit with a reaction transport model to me. What about these data being colloidal fractions that pass the 0.2 _m poresize of the filter that would have little reactivity as shown by the slight slopes that look mostly within error bars? I agree that 1 point from station 8 and two from station 1 are clearly above error bar range but only station 8 seems to indicate that you caught the BBL. However, station 1 profile points to 2 homogeneous layers that suggest diffusion is not dominant there Silica data should appear somewhere*

Response to referee comment 12:

The determination of the removal rates of Fe in near bottom waters are based on the assumption that silica is transported vertically by eddy diffusion and the modelled eddy diffusion coefficient from our data determines the half-life of Fe in the first 4 m above the seafloor. It is possible that our assumption of solute transport by eddy diffusion is not correct. Alternatively, decreasing Fe and Silica concentration above the seafloor could be due to super-imposed water layers with different Fe and silica concentrations but little vertical exchange. In this case our calculated half-life would be an underestimation. We are very aware of these limitations and will expand section 4.1.2 to explicitly discuss the uncertainties of this method.

We only applied the half-life calculations at stations with a discernable gradient within the first 4 m above the seafloor. This is clearly the case at station 4, but even though the gradient at station 3 falls within error we decided to also use this station because it still shows a clear gradient when compared to silica.

The silica concentrations measured in near bottom waters will be listed in a table in the appendix.

Referee comment 13:
*4.2.1. Line 632 is it possible to detail how CdS precipitation in the near bottom water column was calculated? Show linear regressions, statistical significance and determination coefficients. What are the ranges in table 3? This section is very interesting but I feel uncomfortable with the lack of details about calculations and numbers out of it. It is a pity because it is I guess the post important part of the study*

Response to referee comment 13:
The determination of the particulate delivery of Cd to the sediment via CdS precipitation in the water column is explained below table 3. We will also add an explanation to the main text in section 4.2.2 Quantification of the sedimentary Cd sink.

We will plot the linear regressions for the concentration change over time during the benthic chamber incubation for Fe and Cd (figures 5 and 7). We will also mention the coefficient of determination for each linear regression.

The ranges in table 3 listed in column "Cd from organic matter" result from the minimum and maximum estimate of Cd that is delivered to the sediment via organic material (see below table 3 and explanation in section 4.2.2). These values were determined by multiplying the average concentration ratio of Cd to C in phytoplankton (Moore et al., 2013) with published particulate organic carbon rain rates (maximum estimate) or burial rates (minimum estimate) for each individual site (Dale et al., 2015). The ranges in table 3 in the column "CdS precipitation in water column" result from the subtraction of the minimum and maximum of Cd that is delivered by organic matter, we add this information to the explanation below table 3.

Referee comment 14:
*Conclusions Station 1 seems to weaken the first statement of the conclusion The order of TM affinity towards sulfide is shown for the first time in the conclusion. It should be said more explicitly in the introduction*

Response to referee comment 14:
We will add to the introduction in line 60 the order of decreasing sulfide solubility for several mono-sulfide forming trace metals (e.g. Fe>Zn>Cd).

Referee comment 15:
*The expansion of OMZ is evocated to draw a future scenario of TM burial/recycling within the OMZ but what about seasonality and other cyclic controls such as el nino/la nina conditions? Conclusions seem a bit too bold in that way. I am not sur that a putative enhancement or decrease of TM upward flux to surface waters (that is by the way not shown here with the Cd example) would affect marine ecology. At least nothing in the paper discusses it. If I read only the conclusion and the abstract, this will be the take-home message of the paper I would guess and the paper does not address such important question.*

Response to referee comment 15:
In our manuscript we want to focus on long-term environmental changes, even though we are aware that seasonality or non-cyclic oceanographic and meteorological features as ENSO are important to consider. There are other publications dealing with these short term changes e.g. by Scholz et al., 2011, who discussed the impact of ENSO-related oxygen fluctuations on the early diagenesis of redox-sensitive trace metals.

The concentration/reservoir of a trace metal in the ocean is dependent on the respective input and output of this element. Consequently, benthic recycling and burial processes are an important aspect of oceanic mass balance. As especially Fe and also other trace metals can be often the limiting factor for primary productivity the availability of TMs has a direct impact on phytoplankton growth. As bottom water can be upwelled to the surface ocean, the benthic burial or release of a trace metal can have an impact on the amplitude or fuel/limit primary productivity. We will refer to this last statement in the introduction.

Technical referee comment 1:
*Line 189 "compared" Line 191 "covered" Line 207 "weighted" Line 317 what are optopodes? Line 563 "can take"*

Response to technical referee comment 1:
These mistakes will be corrected.

Technical referee comment 2:
*Figures 5 and 7 should show linear regressions and equations as well as error bars for the data points. Then, equations and quality of regressions should be quickly described to prepare the discussion section about iron being oxidized by nitrogen compounds*

Response to technical referee comment 2:

We will implement these suggestions. We will plot linear regressions for the concentration change over time during the benthic chamber incubation for Fe and Cd (figures 5 and 7). We will also mention the coefficient of determination for each linear regression, which will also be briefly described.

Technical referee comment 3:
*Details from saturation indices shown in the caption of figure 10 should maybe appear within the text of the manuscript in the methods or the discussion*

Response to technical referee comment 3:
We list the order of sulfide solubility for different trace metals in the discussion in lines 687 - 689.

Technical referee comment 4:
*Table 3 what is within brackets for station 1 flux from chambers. This calculation would show what evolution of Cd concentration within the chamber? Is that coherent with your data?*

Response to technical referee comment 4:
In table 3 the numbers in brackets represent the flux calculated from the Cd concentration difference between the bottom water (0.5 m) and the first sample from the benthic chamber (taken after 0.25 h) which was characterized by near-complete Cd depletion. This is explained below table 3.

The flux is coherent with our data in the sense that it would be high enough to explain the Cd enrichment in the sediment. We will explain that in section 4.2.2 Quantification of the sedimentary cadmium sink.

Technical referee comment 5:
*Figure 9 reminds me a figure from my paper about cadmium fluxes in a seasonal hypoxic lagoon (Metzger et al., 2007) with a kind of threshold. You could put in the x-axis oxygen concentration and or sulfide and or nitrate. Why did you do only a figure for iron?*

Response to technical referee comment 5:
In figure 9 the spatial extension of biogeochemical conditions are indicated by bars above the figure (oxygenated, nitrogenous, sulfidic).

Indeed, it would be very interesting to compare also the variability of Cd fluxes between different seasons or years under changing biogeochemical conditions. Unfortunately, however, no benthic Cd fluxes were determined or published prior to our 2017 cruise.

References

Dale, A. W., Sommer, S., Lomnitz, U., Montes, I., Treude, T., Liebetrau, V., Gier, J., Hensen, C., Dengler, M., Stolpovsky, K., Bryant, L. D. and Wallmann, K.: Organic carbon production, mineralisation and preservation on the Peruvian margin, Biogeosciences, 12(5), 1537–1559, doi:10.5194/bg-12-1537-2015, 2015.

Lenstra, W. K., Hermans, M., Séguret, M. J. M., Witbaard, R., Behrends, T., Dijkstra, N., van Helmond, N. A. G. M., Kraal, P., Laan, P., Rijkenberg, M. J. A., Severmann, S., Teacă, A. and Slomp, C. P.: The shelf-to-basin iron shuttle in the Black Sea revisited, Chem. Geol., 511(April), 314–341, doi:10.1016/j.chemgeo.2018.10.024, 2019.

Metzger, E., Simonucci, C., Viollier, E., Sarazin, G., Prévot, F., Elbaz-Poulichet, F., Seidel, J.-L. and Jézéquel, D.: Influence of diagenetic processes in Thau lagoon on cadmium behavior and benthic fluxes, Estuar. Coast. Shelf Sci., 72(3), 497–510, doi:10.1016/j.ecss.2006.11.016, 2007.

Moore, C. M., Mills, M. M., Arrigo, K. R., Berman-Frank, I., Bopp, L., Boyd, P. W., Galbraith, E. D., Geider, R. J., Guieu, C., Jaccard, S. L., Jickells, T. D., La Roche, J., Lenton, T. M., Mahowald, N. M., Marañón, E., Marinov, I., Moore, J. K., Nakatsuka, T., Oschlies, A., Saito, M. A., Thingstad, T. F., Tsuda, A. and Ulloa, O.: Processes and patterns of oceanic nutrient limitation, Nat. Geosci., 6(9), 701–710, doi:10.1038/ngeo1765, 2013.

Noffke, A., Hensen, C., Sommer, S., Scholz, F., Bohlen, L., Mosch, T., Graco, M. and Wallmann, K.: Benthic iron and phosphorus fluxes across the Peruvian oxygen minimum zone, Limnol. Oceanogr., 57(3), 851–867, doi:10.4319/lo.2012.57.3.0851, 2012.

Pakhomova, S. V., Hall, P. O. J., Kononets, M. Y., Rozanov, A. G., Tengberg, A. and Vershinin, A. V.: Fluxes of iron and manganese across the sediment–water interface under various redox conditions, Mar. Chem., 107(3), 319–331, doi:10.1016/j.marchem.2007.06.001, 2007.

Point, D., Monperrus, M., Tessier, E., Amouroux, D., Chauvaud, L., Thouzeau, G., Jean, F., Amice, E., Grall, J., Leynaert, A., Clavier, J. and Donard, O. F. X.: Biological control of trace metal and organometal benthic fluxes in a eutrophic lagoon (Thau Lagoon, Mediterranean Sea, France), Estuar. Coast. Shelf Sci., 72(3), 457–471, doi:10.1016/j.ecss.2006.11.013, 2007.

Rapp, I., Schlosser, C., Rusiecka, D., Gledhill, M. and Achterberg, E. P.: Automated preconcentration of Fe, Zn, Cu, Ni, Cd, Pb, Co, and Mn in seawater with analysis using high-resolution sector field inductively-coupled plasma mass spectrometry, Anal. Chim. Acta, 976, 1–13, doi:10.1016/j.aca.2017.05.008, 2017.

Scholz, F., Hensen, C., Noffke, A., Rohde, A., Liebetrau, V. and Wallmann, K.: Early diagenesis of redox-sensitive trace metals in the Peru upwelling area - response to ENSO-related oxygen fluctuations in the water column, Geochim. Cosmochim. Acta, 75(22), 7257–7276, doi:10.1016/j.gca.2011.08.007, 2011.

Scholz, F., Schmidt, M., Hensen, C., Eroglu, S., Geilert, S., Gutjahr, M. and Liebetrau, V.: Shelf-to-basin iron shuttle in the Guaymas Basin, Gulf of California,

Geochim. Cosmochim. Acta, 261, 76–92, doi:10.1016/j.gca.2019.07.006, 2019.

Shibamoto, Y. and Harada, K.: Silicon flux and distribution of biogenic silica in deep-sea sediments in the western North Pacific Ocean, Deep Sea Res. Part I Oceanogr. Res. Pap., 57(2), 163–174, doi:10.1016/j.dsr.2009.10.009, 2010.

Sundby, B., Anderson, L. G., Hall, P. O. J., Iverfeldt, Å., van der Loeff, M. M. R. and Westerlund, S. F. G.: The effect of oxygen on release and uptake of cobalt, manganese, iron and phosphate at the sediment-water interface, Geochim. Cosmochim. Acta, 50(6), 1281–1288, doi:10.1016/0016-7037(86)90411-4, 1986.

Turetta, C., Capodaglio, G., Cairns, W., Rabar, S. and Cescon, P.: Benthic fluxes of trace metals in the lagoon of Venice, Microchem. J., 79(1–2), 149–158, doi:10.1016/j.microc.2004.06.003, 2005.

Warnken, K. W., Gill, G. A., Griffin, L. L. and Santschi, P. H.: Sediment-water exchange of mn, fe, ni and zn in galveston bay, texas, Mar. Chem., 73(3–4), 215–231, doi:10.1016/S0304-4203(00)00108-0, 2001.

---

## Author Comment (AC2) · 17 Feb 2020

Dear Michael Staubwasser, we would like to thank you for reviewing our manuscript and your constructive comments.

In this response letter *referee comments are listed in italic font*, followed by the authors responses to each individual comment. Changes that will be made to the manuscript are underlined.

Referee comment 1:

*The input data required to verify the authors' flux calculations from concentration gradients between the uppermost pore water sample and the water column sample is absent from the manuscript's figures and tables. That includes the porosity used in equation 3 and the tortuosity calculated therefrom in eq. 2. Nor are the input concentration values given. I cannot glean them from fig. 4, as the scale is too small and most pore water profiles appear to show an enormous spread in the uppermost two data points. What values were used for the calculation? Which one is the true surface in fig. 4? What is the reason for this spread in pore water concentrations so close to the surface - e.g. St1_M136 (Fe: ~ 100 nM and >1 nM) and St4_M136 (Fe: low nM range? and > 6 µM). Comparing bottom water column data (fig. 3) does not suggest one of these plotted pore water values could actually be the bottom water concentration. I think it would be better, to calculate concentration gradients from more than just two data points for Fe and Cd. Perhaps the authors would also consider to show the 0-500 nM range in high resolution, and the few numbers above at lower resolution with a break in the axis between?*

Response to referee comment 1:

We fully agree that the input data for the diffusive flux calculations need to be listed in the manuscript. The input data (i.e., porosity, concentration values, in-situ temperature, pressure and salinity) will be listed in a table in the appendix.

For the diffusive flux calculations we deliberately decided to use the two point concentration gradient between the uppermost pore water sample and the overlying bottom water. We know that this method has its limitations, but need to stay with this approach for the following reasons.

1. This is a commonly used approach which allows us to compare our diffusive fluxes to fluxes from other publications (e.g. Sundby et al., 1986; Warnken et al., 2001; Turetta et al., 2005; Pakhomova et al., 2007; Noffke et al., 2012; Lenstra et al., 2019).

2. The strong gradients between the uppermost pore water and bottom water sample is typically observed within open-marine oxygen minimum zones (e.g. Noffke et al., 2012; Scholz et al., 2019). Since $H_2S$ accumulates close to the surface in these settings, the dissolved Fe peak in pore water is very narrow (1 – 2 cm) and located close to close to the sediment-water interface. Applying more advanced methods for the determination of diffusive fluxes, e.g. curve fitting, would fail to capture the sharp

gradients at the sediment surface and, thus, lead to erroneous flux estimates (see also Shibamoto and Harada, 2010; Dale et al., in prep.)

3. Bottom water concentrations are very small (maximum a few hundreds of nM) compared to the concentrations in the uppermost pore water sample (several µM). Therefore, the gradient between pore water and the bottom water is mostly dependent on the pore water concentration. For the same reason, varying the bottom water concentration over the range observed in our data set has a negligible effect on the benthic flux.

We will explicitly include the above explanation in section 2.3 Flux calculations.

Referee comment 2:

*I also don't like the way these gradient-based diffusive flux calculations (lines) are shown over the flux chamber time series data in fig. 5. Better use the water column data as a starting point - since chambers were flushed with ambient bottom water - than the first chamber time series value. There often is quite a difference between the ambient bottom water and the first chamber value that the authors do not explain. Sometimes that difference is hard to understand. In Sta 1 ambient water is higher than the starting value of the chamber (flushed with ambient water). At Sta 9, chamber values (~ 80 nM) are much higher throughout than ambient water (~ 5 nM), but apparently there is no diffusive flux of Fe. That discrepancy does not agree with the authors' general statement in the abstract that the two flux estimates agree with each other.*

Response to referee comment 2:

Different Fe concentrations inside and outside of the chamber are to be expected because of diffusive Fe release from the sediment and an accumulation in the enclosed water volume inside the benthic chamber. Furthermore, the bottom water enclosed in the benthic chamber is much closer to the seafloor compared to the bottom water samples collected closest to the seafloor (at 0.5 m). We assume that the concentrations in the bottom water enclosed in the chamber represents the average concentration over the first 30 cm above the seafloor. Outside of the chamber, a concentration gradient between the seafloor and the bottom-near water column is likely to establish because of diffusive release of Fe from the sediment and removal in the bottom water. Therefore, it is not surprising that different concentrations are observed in the chamber and in the bottom water 50 cm above the seafloor.

Furthermore, the water sample at 0.5 m away from the seafloor was collected in sampling bags with peristaltic pumps over a time span of 32 hours. This means that, in contrast to the benthic chamber, where samples were taken within minutes at discrete points in time, short-term concentration changes cannot be seen.

For the reasons given above, we cannot expect to see the same concentration value in different sample types and cannot use bottom water values as a starting point for the incubations.

The above explanations will be added to the discussion in section 4.1.1 Comparison of diffusive and in-situ benthic chamber iron fluxes.

We will add linear regressions for the concentration change over time during the benthic chamber incubation (figures 5 and 7). We prefer to also present the slopes calculated from pore water profiles to facilitate the comparison of benthic fluxes derived by these two different approaches.

Referee comment 3:

*Generally, I see an unaddressed problem in that the Fe concentration data for the near-bottom water column and the flux chambers were filtered at 0.2 µm and acidified for later analysis. That would dissolve all colloidal Fe, which I would expect to be quite abundant near the bottom (see e.g. Fitzsimmons et al. 2015, Mar Chem 173 pp 125 for a similar problem in ocean water underneath dust plumes, and Heller et al. 2017, GCA 211, pp 174 for a discussion of particulate - and colloidal - removal of dFe in the Peruvian OMZ). I am aware of the authors citing the latter manuscript, but they do not cite it in the context of colloidal/particulate removal of dFe. I realize, that the necessary ultra-filtration to quantify colloidal Fe was not implemented, and of course, cannot be provided in hindsight. However, I would like to ask the authors to deal with this problem in the discussion. There is so much scatter in the Fe concentration time series from the flux chambers, that I am certain the dFe data suffer from colloids. Colloidal Fe would not count for flux calculations based on pore water - bottom water concentration gradients, and would very much complicate flux estimates based on the linear trend calculation from the flux chamber time series. My quick suggestion would be to provide two estimated for fluxes based on minimum and maximum estimates, but perhaps the authors can come up with a better way to deal with this issue. A discussion of this aspect would really benefit the manuscript. I suggest to discuss this aspect at the end of paragraph 4.1.1. In addition, bioirrigation should contribute considerably to resuspension of particles and colloids as well - a process which the authors briefly mention in lines 441-446 in general terms but do not seem to connect to their scattered flux chamber data.*

Response to referee comment 3:

We agree that colloidal or nanoparticulate Fe are likely to play a role in Fe cycling on the Peruvian margin, especially close to seafloor, where particles are quite abundant. Unfortunately, we did not differentiate colloidal and truly dissolved fractions during our sampling, which is why we cannot discuss this aspect based on our data.

The transfer of Fe between dissolved, colloidal and particulate pools is likely to affect the balance between dissolved Fe transport away from the seafloor or re-precipitation and deposition to some extent. To outline the role of suspended particles in modulating benthic Fe fluxes we already refer to the work of Homoky et al., 2012. In addition, we will take the following steps to address the reviewer's comment:

We will make it clearer in section 4.1.1 that colloidal Fe could modify Fe concentrations within our samples, as colloids are quite reactive. They are much more soluble than larger particles and can be rapidly reduced and dissolved in anoxic environments, but

they can also be an intermediate step in dissolved Fe removal through aggregation of larger particles (Raiswell and Canfield, 2012). On the other hand, Fitzsimmons and Boyle, 2014 observed that soluble (< 0.02 μm) rather than colloidal Fe was the dominant fraction within the oxygen minimum zone in the tropical north Atlantic.

We will refer to the processes mentioned above to explain the scatter in Fe concentrations observed during benthic chamber incubations.

We don't understand the reference to Heller et al., 2017 in the context of colloids as this publication does not discuss the role of colloidal Fe.

Finally, we fully agree with the reviewer's comment that further research on particle dissolved interactions is needed and we will explicitly mention this in the revised version (section 4.1.1. Comparison of diffusive and in-situ benthic chamber iron fluxes, starting from line 488).

We do connect our scattered flux chamber data at stations 1, 9 and 10 to bioturbation and bioirrigation in lines 433 – 436, we will add to this statement, that bioturbation and bioirrigation could also lead to particle or colloid resuspension.

Referee comment 4:

*In the abstract, lines 20-23, I would suggest the authors phrase the suggested "agreement" of different flux calculations somewhat more careful. This is not a very convincing statement even without considering the colloid problem. At best, there is agreement for some of the stations.*

Response to referee comment 4:

Following the reviewer's recommendation, we will rephrase lines 20 – 23 to make clear that we distinguish between Fe fluxes within the permanent oxygen minimum zone and outside of the permanent oxygen minimum zone. The benthic diffusive Fe fluxes and Fe fluxes from benthic chamber incubations do mostly agree well within the permanent oxygen minimum zone, whereas outside of the oxygen minimum zone there is a larger discrepancy between the two estimates.

Referee comment 5:

*In the conclusion, I don't think it is appropriate to claim that Fe just diffuses out of the sediment column. The authors themselves have made the case for biorrigation and bioturbation as a significant process for Fe enrichment in the flux chambers.*

Response to referee comment 5:

We will rephrase this part of the conclusions and only refer to diffusion as being the dominant process for Fe fluxes within the permanent oxygen minimum zone, where bottom-dwelling macrofauna is absent. We are aware that additional processes could

play a role within the permanent oxygen minimum zone. However, we feel it is appropriate to only refer to the main process in the conclusions.

Technical referee comments:

*Technical aspects that should be corrected: a) Please show cruise AND station number in figures 1 and 2. b) I believe it would be useful to show the H2S data along with the Cd data in fig. 6, since they are discussed together.*

Response to technical referee comments:

We will implement both your suggestions a) and b). Station numbers will be added to the map in fig. 1 and to the caption of fig. 2. We will display the profiles of $H_2S$ concentrations in pore waters not only in figure 4 but also in figure 6.

References

Fitzsimmons, J. N. and Boyle, E. A.: Both soluble and colloidal iron phases control dissolved iron variability in the tropical North Atlantic Ocean, Geochim. Cosmochim. Acta, 125, 539–550, doi:10.1016/j.gca.2013.10.032, 2014.

Heller, M. I., Lam, P. J., Moffett, J. W., Till, C. P., Lee, J. M., Toner, B. M. and Marcus, M. A.: Accumulation of Fe oxyhydroxides in the Peruvian oxygen deficient zone implies non-oxygen dependent Fe oxidation, Geochim. Cosmochim. Acta, 211, 174–193, doi:10.1016/j.gca.2017.05.019, 2017.

Homoky, W. B., Severmann, S., McManus, J., Berelson, W. M., Riedel, T. E., Statham, P. J. and Mills, R. A.: Dissolved oxygen and suspended particles regulate the benthic flux of iron from continental margins, Mar. Chem., 134–135, 59–70, doi:10.1016/j.marchem.2012.03.003, 2012.

Lenstra, W. K., Hermans, M., Séguret, M. J. M., Witbaard, R., Behrends, T., Dijkstra, N., van Helmond, N. A. G. M., Kraal, P., Laan, P., Rijkenberg, M. J. A., Severmann, S., Teacă, A. and Slomp, C. P.: The shelf-to-basin iron shuttle in the Black Sea revisited, Chem. Geol., 511(April), 314–341, doi:10.1016/j.chemgeo.2018.10.024, 2019.

Noffke, A., Hensen, C., Sommer, S., Scholz, F., Bohlen, L., Mosch, T., Graco, M. and Wallmann, K.: Benthic iron and phosphorus fluxes across the Peruvian oxygen minimum zone, Limnol. Oceanogr., 57(3), 851–867, doi:10.4319/lo.2012.57.3.0851, 2012.

Pakhomova, S. V., Hall, P. O. J., Kononets, M. Y., Rozanov, A. G., Tengberg, A. and Vershinin, A. V.: Fluxes of iron and manganese across the sediment–water interface under various redox conditions, Mar. Chem., 107(3), 319–331, doi:10.1016/j.marchem.2007.06.001, 2007.

Raiswell, R. and Canfield, D. E.: The Iron Biogeochemical Cycle Past and Present,

Geochemical Perspect., 1(1), 1–220, doi:10.7185/geochempersp.1.1, 2012.

Scholz, F., Schmidt, M., Hensen, C., Eroglu, S., Geilert, S., Gutjahr, M. and Liebetrau, V.: Shelf-to-basin iron shuttle in the Guaymas Basin, Gulf of California, Geochim. Cosmochim. Acta, 261, 76–92, doi:10.1016/j.gca.2019.07.006, 2019.

Shibamoto, Y. and Harada, K.: Silicon flux and distribution of biogenic silica in deep-sea sediments in the western North Pacific Ocean, Deep Sea Res. Part I Oceanogr. Res. Pap., 57(2), 163–174, doi:10.1016/j.dsr.2009.10.009, 2010.

Sundby, B., Anderson, L. G., Hall, P. O. J., Iverfeldt, Å., van der Loeff, M. M. R. and Westerlund, S. F. G.: The effect of oxygen on release and uptake of cobalt, manganese, iron and phosphate at the sediment-water interface, Geochim. Cosmochim. Acta, 50(6), 1281–1288, doi:10.1016/0016-7037(86)90411-4, 1986.

Turetta, C., Capodaglio, G., Cairns, W., Rabar, S. and Cescon, P.: Benthic fluxes of trace metals in the lagoon of Venice, Microchem. J., 79(1–2), 149–158, doi:10.1016/j.microc.2004.06.003, 2005.

Warnken, K. W., Gill, G. A., Griffin, L. L. and Santschi, P. H.: Sediment-water exchange of mn, fe, ni and zn in galveston bay, texas, Mar. Chem., 73(3–4), 215–231, doi:10.1016/S0304-4203(00)00108-0, 2001.

---

## Author Response (AR1)

We would like to thank Edouard Metzger and Michael Staubwasser for their constructive reviews.

In this response, *referee comments are listed in italic font*, followed by the authors responses to each individual comment. Changes that were made to the manuscript are underlined. The revised manuscript including track changes is listed below.

**Referee 1 comment 1:**

*Abstract L41-42 I am not sure to follow the causality between metal relative metal solubility and spatial-temporal heterogeneity L45-47 the last sentence of the abstract tends to suggest that decrease of cadmium solubility due to sulfide increase consequently to oxygen declining may affect marine ecosystems. The sentence appears to me overstated, out of the scope of what your data can say, overall, as a general sentence that would appeal to emotion. I would suggest to find another sentence to take perspective. Maybe adding another metal that you must have analyzed usinc a ICPMS to the study showing different sulfide affinities would have helped to strengthen such a statement.*

**Response to referee 1 comment 1:**

We are using the elements Fe and Cd as prototypes, as they have a contrasting sulfide solubility, and interpolate their behavior to trace metals which have an intermediate sulfide solubility (lines 86 – 92, 686 – 689, 699 - 701). The concentration/reservoir of a trace metal in the ocean is dependent on the respective input and output fluxes of this element. Consequently, benthic recycling and burial processes are an important aspect of the oceanic mass balance. As especially Fe and also other trace metals can often be the limiting factor for primary productivity, the availability of trace metals has a direct impact on phytoplankton growth. Where bottom water is upwelled to the surface ocean, the benthic burial or release of a trace metal can limit or fuel primary productivity.

We added an explanation to make this causality clearer in the revised version in lines 88 – 103 "Because of their contrasting tendency to form sulfide minerals and different supply pathways to the sediment, Fe and Cd can serve as prototypes to provide information about how sedimentary fluxes of different TMs may respond to declining oxygen concentrations. Under more reducing conditions the mobility of TMs can either be enhanced or diminished, e.g., through precipitation of sulfide minerals that are buried in the sediments (e.g. Westerlund et al., 1986; Rigaud et al., 2013; Olson et al., 2017). Increased burial or release of TMs at the seafloor can have an impact on the amplitude of primary productivity, especially at the eastern ocean boundaries where the near-bottom water column is connected to the surface ocean via upwelling. Moreover, since the inventories of TMs in the ocean are generally dependent on the respective input and output fluxes, changes in the balance between trace metal recycling and burial can have an impact on oceanic TM reservoirs on longer timescales. By comparing the benthic biogeochemical cycling of Fe and Cd across spatial and temporal redox gradients, we aim to provide general constraints on how the stoichiometry of bio-essential TMs in seawater may be affected by ocean deoxygenation.".

We are currently working on a second paper where we will present aspects on sedimentary cycling and benthic fluxes of other trace metals off Peru including Zn, which has a low sulfide solubility and Co which has a high sulfide solubility, that are in agreement with the conclusions of the present manuscript.

**Referee 1 comment 2:**
*Introduction All along the introduction, it is stated that little is known about TM benthic fluxes in hypoxic environments. I would disagree with that. The authors cite themselves several of them and I think of studies done by Faganeli and coworkers in the Adriatics, work done during the Microbent project in Thau lagoon including by myself. Work in hypoxic estuaries such as the Gironde or the Scheldt among others. I have the feeling that if authors had detailed more about TM behavior in different environments during occurrence of hypoxia/anoxia and sulfide release into the water column they could have made a series of predictive hypotheses on what would be Cd behavior in the Peruvian OMZ. Overall, I am not a fan of the outline of the introduction because the rationale is at the beginning and the state of the art shown after is too far to make the reader quite sur about what are really the hypotheses made here and what is really new.*

**Response to referee 1 comment 2:**
It was not our purpose to disregard earlier work on sedimentary trace metal dynamics in inner-shore and coastal environments like lagoons and estuaries. In sections 1.2 and 1.3 we refer to earlier studies on benthic trace metal biogeochemistry. We note, however, that except for Fe, little work has been done in open marine systems where the redox- and sediment-dynamics are different compared to estuaries and lagoons. The expansion of our knowledge on trace metal cycling and fluxes in these open-marine environments is the goal of our study.

In the revised manuscript we included further citations to studies in inner-shore environments such as  Metzger et al., 2007 and Point et al., 2007 and rephrased lines 155 – 160 "Most previous studies have focused on the benthic cycling of Cd in near- and in-shore environments such as estuaries and lagoons (e.g. Westerlund et al., 1986; Colbert et al., 2001; Audry et al., 2006b; Metzger et al., 2007; Point et al., 2007; Scholz and Neumann, 2007). By contrast, little is known about Cd cycling in open-marine sedimentary environments, where the redox- and sediment-dynamics are different.".

In the general section on trace metal behavior (1.1 Scientific rationale) we refer now more explicitly to the knowledge gained in studies on estuaries and lagoons that trace metals can precipitate as sulfides when redox conditions change and $H_2S$ concentrations increase in the revised version in lines 91 -94 " Under more reducing conditions the mobility of TMs can either be enhanced or diminished, e.g., through precipitation of sulfide minerals that are buried in the sediments (e.g. Westerlund et al., 1986; Rigaud et al., 2013; Olson et al., 2017).".

**Referee 1 comment 3:**
*Methods Incubation time was of 32h, how oxygen evolved within chambers that were not anoxic in the beginning?*

**Response to referee 1 comment 3:**
The evolution of oxygen concentrations during the benthic lander incubations will be published by Clemens et al. (in preparation).

The incubated bottom water in chambers below the oxygen minimum zone (station 9 and 10) did not become anoxic. At station 9 (750 m) the oxygen concentrations decreased from around 3 µmol at the beginning of the incubation to around 2 µmol at the end of the incubation. At the station 10 (950 m) oxygen remained above 10 µmol throughout the incubation time. For the incubation at station 1 (75 m) we cannot exclude that traces of oxygen were present during the first 5 hours of the incubation (some air bubbles may have been trapped inside the chamber during the lowering through the water column, that then dissolved during the beginning of the incubation). This may have reduced dissolved Fe concentrations in the first sample of this incubation through oxidative removal, however excluding this first value would not affect the overall flux magnitude derived from this incubation.

**Referee 1 comment 4:**
*How the benthic boundary layer's thickness is established and what is its dynamics? That should be mentioned in the introduction as how it may affect benthic fluxes What was the detection and quantification limit for Cd measurements*

**Response to referee 1 comment 4:**
It was our goal to sample and to capture the cycling of trace metal cycling in the highly dynamic layer overlying the seafloor. The common definition of the benthic boundary layer is "a discrete layer of flowing sea water above a benthic substrate, delimited vertically by its contact with free stream flow" (D.J. Wildish, in Encyclopedia of Ocean Sciences (Second Edition), 2001). We acknowledge that, strictly speaking, our sampling interval does not correspond to the benthic boundary layer in the proper sense. We therefore decided to refer to "near-bottom water column" in the revised version.

For Cd measurements we applied the method by Rapp et al., 2017, where the detection limit for Cd is 0.8 pmol L$^{-1}$, which is well below the lowest concentrations of our samples (40 pmol L-1). The detection limit of Cd is listed now in lines 293 - 294 in the revised version "The detection limits were 28.8 pM for Fe and 0.8 pM for Cd (Rapp et al., 2017).".

**Referee 1 comment 5:**
*I am aware that there is plenty of literature with diffusive flux calculations from overlying water and the first porewater concentrations. I think this is wrong as soon as a 2-point calculation is subject to the precision of those points and overlying water concentration is an average of 10 to 30 cm of water column according to the fullness of the core. If any author took the entire porewater profile to make an averaged concentration for*

*gradient calculation nobody would take the calculation seriously. I would at least be skeptical about calculations taking only two points to model a line from which, one is an average of something it is impossible to fully describe with such sampling method. I would say that this can be overtaken adding a supplementary point for the gradient determination. Unfortunately, only one of your profiles can apply. This aspect must be clearly discussed in the manuscript and conclusions made from such calculations carefully done. High resolution methods exist.*

**Response to referee 1 comment 5:**
For the diffusive flux calculations we deliberately decided to use the two point concentration gradient between the uppermost pore water sample and the overlying bottom water. We know that this method has its limitations, but need to stay with this approach for the following reasons.

1. This is a commonly used approach which allows us to compare our diffusive fluxes to fluxes from other publications (e.g. Sundby et al., 1986; Warnken et al., 2001; Turetta et al., 2005; Pakhomova et al., 2007; Noffke et al., 2012; Lenstra et al., 2019).

2. The strong gradients between the uppermost pore water and bottom water sample is typically observed within open-marine oxygen minimum zones (e.g. Noffke et al., 2012; Scholz et al., 2019). Since $H_2S$ accumulates close to the surface in these settings, the dissolved Fe peak in pore water is very narrow (order of 1 - 2 cm) and located close to the sediment-water interface. More advanced curve fitting methods would fail to capture sharp gradients at the sediment surface and would, therefore, yield erroneous flux estimates (see also Shibamoto and Harada, 2010; Dale et al., in prep.)

We included an explanation for choosing the two point concentration gradient for diffusive flux calculations in lines 321 – 328 in the revised version "We chose to use the commonly applied approach of a two point concentration gradient for the determination of diffusive fluxes, as more advanced curve fitting methods would fail to capture sharp concentration gradients at the sediment surface and, thus, lead to erroneous flux estimates (Shibamoto and Harada, 2010).".

**Referee 1 comment 6:**
*Linear regressions with standard deviation of the slope are not shown in the document, this should appear in the graphics with also error bars for each point*

**Response to referee 1 comment 6:**
We plotted linear regressions for the concentration change over time during the benthic chamber incubation for Fe and Cd (Fig. 5 and 7). We also listed the equations and the coefficient of determination for each linear regression in the supplement in Table S4.

In figures where error bars are not shown, the analytical error is smaller than the symbol size, which is stated in the figure captions.

**Referee 1 comment 7:**

*Particulate flux calculations and CdS water column uptake calculations should also be detailed here, especially the last one*

**Response to referee comment 7:**
The calculations used for the quantification of the sedimentary Cd sink and the different delivery pathways to the sediment are explained in section 4.2.2 Quantification of the sedimentary Cd sink. How we determine the particulate delivery of Cd to the sediment via CdS precipitation in the near-bottom water column is explained below Table 3 (Cd enrichment in the sediment after subtraction of diffusive and organic Cd sources). We explain this better in lines 676 – 679 in the revised version "The Cd delivery via precipitation in the water column was determined as the remainder of Cdxs * MAR after subtraction of the two other sources (i.e., diffusive flux and minimum/maximum delivery by organic material).".

**Referee 1 comment 8:**
*I think that section 3.1. appeals a lot to literature to be part of the result section. Only bottom oxygen data is described from figure 2. I woud suggest authors to discard the figure 2, table 2 is sufficient, and put that description in the methods sections For a result section I would avoid naming sections using processes such a "biogeochemical cycling". I would suggest to simply call it: "Porewater and benthic fluxes" then "iron, then "cadmium"*

**Response to referee 1 comment 8:**
In section 3.1 we present our measured data of oxygen, nitrate and nitrite during both cruises. The prevailing biogeochemical conditions are important for the trace metal cycling- That is why it is crucial to leave this data in the manuscript.

Regarding the very special oceanic condition, the occurrence of a coastal El Niño during our cruise in 2017, we refer to literature instead of our own data as this includes atmospheric and oceanographic observations that we could not conduct ourselves. We moved this part to the introduction to section 1.4 Study area.

We fully agree and changed the titles of the results sections as suggested in 3.2 Bottom water, pore water and benthic flux data and then in subsections 3.2.1 Iron and 3.2.2 Cadmium.

**Referee 1 comment 9:**
*I would also suggest to avoid citations. Why citations are provided for iron and not for cadmium? Some symmetry should be maintained between these elements at least here to underline differences in results*

**Response to referee 1 comment 9:**
We removed the first part of the sentence starting in line 354 including the citations in line 355 -356.

**Referee 1 comment 10:**

*In the methods section I made a comment about the thickness of the benthic boundary layer. According to the 4 metres profile from the bottom, it appears that the benthic boundary layer is clearly thinner than 0.5 m as concentrations are stable in your date for almost all profiles. I believe from in situ data of oxygen profiles that this boundary layer is within the range of few micrometers to few centimeters. I am not sure those data are really relevant here and could be discarded or remain as supplementary data. Line 351 authors say that initial incubation concentrations were higher than "bottom samples". This pleads to the fact that those bottom samples did not describe the benthic boundary layer. This should be said somehow Did the authors thought of showing modelled gradients in the porewater profiles instead of showing modelled incubation points from diffusive fluxes?*

**Response to referee 1 comment 10:**

As mentioned in response to referee comment 4 we will replace the term 'benthic boundary layer' by the 'near-bottom water column'.

Regarding our statement in line 351 (comparing Fe concentrations in the bottom waters collected inside and outside of the benthic chambers): Different Fe concentrations inside and outside of the chamber are to be expected because of diffusive Fe release from the sediment and an accumulation in the enclosed water volume inside the benthic chamber. Furthermore, the bottom water enclosed in the benthic chamber is much closer to the seafloor compared to the bottom water samples collected closest to the seafloor (at 0.5 m). We assume that the concentrations in the bottom water enclosed in the chamber represents the average concentration over the first 30 cm above the seafloor. Outside of the chamber, a concentration gradient between the seafloor and the bottom-near water column is likely to establish because of diffusive release of Fe from the sediment and removal in the bottom water. Therefore, it is not surprising that different concentrations are observed in the chamber and in the bottom water 50 cm above the seafloor.

Furthermore, the water sample at 0.5 m away from the seafloor was collected in sampling bags with peristaltic pumps over a time span of 32 hours. This means that, in contrast to the benthic chamber, where samples were taken within minutes at discrete points in time, short-term concentration changes cannot be seen.

We added an explanation in lines 432 – 435 in the revised version explaining the different concentrations between bottom water outside and within the benthic chamber "Concentrations of Fe in bottom waters from benthic chamber incubations are mostly higher than in ambient bottom waters because of Fe release from the sediment and an accumulation in the enclosed water volume inside the benthic chamber.".

As mentioned in response to referee comment 6 we plotted linear regressions for the concentration change over time during the benthic chamber incubation for Fe and Cd (figures 5 and 7) together with the slopes calculated from pore water profiles to facilitate the comparison of benthic fluxes derived by these two different approaches.

However, applying a transport-reaction model, which would need to involve carbon, sulfur, nitrogen and iron turnover to generate accurate pore water profiles of dissolved Fe, is beyond the scope of the present paper.

**Referee 1 comment 11:**
*4.1.1. Talking about incubations 1, 9 and 10 you mention bioturbation as a potential artefact, what about oxygen consumption and sulfide precipitation during incubation? Actually a combination of all that could happen as oxygen depletion could bring endo-bionts to surface (see riedel et al experiments in the adriatics) or enhance bioirrigation (Duport et al, 2007) favoring sulfide efflux or iron precipitation. Did you consider leaks in the chamber or dysfunctional homogenization? This is a current issue with in situ incubation. Oxygen and sulphide data from incubations are really missing here to state about ventilation processes or leaking*

**Response to referee 1 comment 11:**
Hydrogen sulfide could not be detected within bottom waters from benthic chamber incubations with our method, which has a detection limit of 1 µmol L$^{-1}$. For Fe to precipitate as sulfide several hundred µmol L$^{-1}$ of H$_2$S are required. Therefore, we exclude Fe sulfide precipitation within the benthic chambers.

As mentioned in our response to referee comment 3, we cannot exclude a slight oxygen contamination within the benthic chamber at station 1 during the very beginning of the incubation (some air bubbles may have been trapped inside the chamber during the lowering trough the water column, that then dissolved during the beginning of the incubation). This may have reduced dissolved Fe concentrations in the first sample of this incubation through oxidative removal. However, excluding this first value would not affect the overall flux magnitude derived from this incubation.

At stations below the permanent oxygen minimum zone (station 9 and 10) the oxygen was not consumed during the incubations, which is why it is unlikely that endobionts came to the sediment surface.

**Referee 1 comment 12:**
*4.1.2. I am not convinced about the half-life calculations since there is little change in iron concentrations and they do not fit with a reaction transport model to me. What about these data being colloidal fractions that pass the 0.2 _m poresize of the filter that would have little reactivity as shown by the slight slopes that look mostly within error bars? I agree that 1 point from station 8 and two from station 1 are clearly above error bar range but only station 8 seems to indicate that you caught the BBL. However, station 1 profile points to 2 homogeneous layers that suggest diffusion is not dominant there Silica data should appear somewhere*

**Response to referee 1 comment 12:**
The determination of the removal rates of Fe in near bottom waters are based on the assumption that silica is transported vertically by eddy diffusion and the modelled eddy diffusion coefficient from our data determines the half-life of Fe in the first 4 m above the seafloor. It is possible that our assumption of solute transport by eddy diffusion is not correct. Alternatively, decreasing Fe and Silica concentration above the seafloor could be due to super-imposed water layers with different Fe and silica concentrations but little vertical exchange. In this case our calculated half-life would be an underestimation. We are very aware of these limitations and expanded section 4.1.2

to explicitly discuss the uncertainties of this method in line 533 – 540 in the revised version "The approach assumes that $Si(OH)_4$ is transported vertically by eddy diffusion and eddy diffusion and oxidation control the half-life of Fe in the first 4 m above the seafloor. It is possible that our assumption of solute transport by eddy diffusion is not correct. Alternatively, decreasing Fe and $Si(OH)_4$ concentration above the seafloor could be due to super-imposed water layers with different Fe and $Si(OH)_4$ concentrations but little vertical exchange. In this case our calculated half-life would be an underestimation.".

We only applied the half-life calculations at stations with a discernable gradient within the first 4 m above the seafloor. This is clearly the case at station 4, but even though the gradient at station 3 falls within error we decided to also use this station because it still shows a clear gradient when compared to silica.

The silica concentrations measured in near bottom waters are listed now in Table S3 in the supplement.

**Referee 1 comment 13:**
*4.2.1. Line 632 is it possible to detail how CdS precipitation in the near bottom water column was calculated? Show linear regressions, statistical significance and determination coefficients. What are the ranges in table 3? This section is very interesting but I feel uncomfortable with the lack of details about calculations and numbers out of it. It is a pity because it is I guess the post important part of the study*

**Response to referee 1 comment 13:**
The determination of the particulate delivery of Cd to the sediment via CdS precipitation in the water column is explained below Table 3. We also explained this better in lines 676 – 679 in the revised version "The Cd delivery via precipitation in the water column was determined as the remainder of $Cd_{xs}$ * MAR after subtraction of the two other sources (i.e., diffusive flux and minimum/maximum delivery by organic material).".

We plotted the linear regressions for the concentration change over time during the benthic chamber incubation for Fe and Cd (figures 5 and 7). We mention now the coefficient of determination for each linear regression in Table S4 in the supplement.

The ranges in Table 3 listed in column "Cd from organic matter" result from the minimum and maximum estimate of Cd that is delivered to the sediment via organic material (see below Table 3 and explanation in section 4.2.2). These values were determined by multiplying the average concentration ratio of Cd to C in phytoplankton (Moore et al., 2013) with published particulate organic carbon rain rates (maximum estimate) or burial rates (minimum estimate) for each individual site (Dale et al., 2015). The ranges in Table 3 in the column "CdS precipitation in water column" result from the subtraction of the minimum and maximum of Cd that is delivered by organic matter, we explain this better now below Table 3.

**Referee 1 comment 14:**

*Conclusions Station 1 seems to weaken the first statement of the conclusion The order of TM affinity towards sulfide is shown for the first time in the conclusion. It should be said more explicitly in the introduction*

**Response to referee 1 comment 14:**
We added to the introduction the order of decreasing sulfide solubility for several mono-sulfide forming trace metals in the revised version in lines 60 – 61 "(e.g. Fe, Zn, Cd; listed in the order of decreasing sulfide solubility).".

**Referee 1 comment 15:**
*The expansion of OMZ is evocated to draw a future scenario of TM burial/recycling within the OMZ but what about seasonality and other cyclic controls such as el nino/la nina conditions? Conclusions seem a bit too bold in that way. I am not sur that a putative enhancement or decrease of TM upward flux to surface waters (that is by the way not shown here with the Cd example) would affect marine ecology. At least nothing in the paper discusses it. If I read only the conclusion and the abstract, this will be the take-home message of the paper I would guess and the paper does not address such important question.*

**Response to referee 1 comment 15:**
In our manuscript we want to focus on long-term environmental changes, even though we are aware that seasonality or non-cyclic oceanographic and meteorological features as ENSO are important to consider. There are other publications dealing with these short term changes e.g. by Scholz et al., 2011, who discussed the impact of ENSO-related oxygen fluctuations on the early diagenesis of redox-sensitive trace metals.

The concentration/reservoir of a trace metal in the ocean is dependent on the respective input and output of this element. Consequently, benthic recycling and burial processes are an important aspect of oceanic mass balance. As especially Fe and also other trace metals can be often the limiting factor for primary productivity the availability of TMs has a direct impact on phytoplankton growth. As bottom water can be upwelled to the surface ocean, the benthic burial or release of a trace metal can have an impact on the amplitude or fuel/limit primary productivity. We refer to this last statement in the introduction in the revised version in lines 94 – 97 "Increased burial or release of TMs at the seafloor can have an impact on the amplitude of primary productivity, especially at the eastern ocean boundaries where the near-bottom water column is connected to the surface ocean via upwelling.".

**Technical comment 1 referee 1:**
*Line 189 "compared" Line 191 "covered" Line 207 "weighted" Line 317 what are optopodes? Line 563 "can take"*

**Response to technical comment referee 1:**
These mistakes were corrected.

**Technical comment 2 referee 1:**
*Figures 5 and 7 should show linear regressions and equations as well as error bars for the data points. Then, equations and quality of regressions should be quickly described to prepare the discussion section about iron being oxidized by nitrogen compounds*

**Response to technical comment 2 referee 1:**
We implemented these suggestions. We plotted linear regressions for the concentration change over time during the benthic chamber incubation for Fe and Cd (Fig. 5 and 7). We also mention the coefficient of determination for each linear regression in the supplement in Table S4. $R^2$ values are briefly described in the revised version in lines 392 - 395 "At these stations also the highest R2 for the linear regressions of the concentration change over the incubation time were calculated (Station 4: R2 = 0.7, Station 6: R2 = 0.5) (Table S4). " and in lines 421 - 423 "At sites within the OMZ, Cd removal within the chamber was near-linear (Stations 4, 5 and 6: $R^2 = \geq 0.9$) (Table S4), which translates to a removal flux of 13 – 23 µmol $m^{-2}$ $y^{-1}$. ".

**Technical comment 3 referee1:**
*Details from saturation indices shown in the caption of figure 10 should maybe appear within the text of the manuscript in the methods or the discussion*

**Response to technical comment 3 referee 1:**
We noticed that the equilibrium constants that were applied were missing and added them to the caption of Fig. 10 in the revised version in lines 1215 – 1218 "Equilibrium constants (log K under standard conditions) for Fe (FeS ppt: -3.92), Ni (millerite: -8.04), Zn (sphalerite: -11.62) and Cd (greenokite: -15.93) were taken from the PHREEQC WATEQ4F database (Ball and Nordstrom, 1991)".

We mention how we calculated the saturation indices in lines 690 -694 "This notion is illustrated in Fig. 10, showing saturation indices calculated based on the range of TM concentrations observed in the ocean and typical $H_2S$ concentrations observed in anoxic marine environments (nM – µM concentrations represent sulfidic events in the water column; µM – mM concentrations are typical for pore waters).".

**Technical comment 4 referee 1:**
*Table 3 what is within brackets for station 1 flux from chambers. This calculation would show what evolution of Cd concentration within the chamber? Is that coherent with your data?*

**Response to technical comment 4 referee:**
In Table 3 the numbers in brackets represent the flux calculated from the Cd concentration difference between the bottom water (0.5 m) and the first sample from the benthic chamber (taken after 0.25 h) which was characterized by near-complete Cd depletion. This is explained below Table 3.

The flux is coherent with our data in the sense that it would be high enough to explain the Cd enrichment in the sediment. We added this statement in the revised version in lines 703 – 707 " At Station 1, where the surface sediment below the benthic chamber was highly sulfidic, the particulate Cd removal calculated from the concentration difference between the bottom water (0.5 m) and the first sample from the benthic chamber incubation (taken after 0.25 h) was high enough to explain the total Cd enrichment in the sediment.".

**Technical comment 5 referee 1:**
*Figure 9 reminds me a figure from my paper about cadmium fluxes in a seasonal hypoxic lagoon (Metzger et al., 2007) with a kind of threshold. You could put in the x-axis oxygen concentration and or sulfide and or nitrate. Why did you do only a figure for iron?*

**Response to technical comment 5 referee 1:**
In figure 9 the spatial extension of biogeochemical conditions are indicated by bars above the figure (oxygenated, nitrogenous, sulfidic).

Indeed, it would be very interesting to compare also the variability of Cd fluxes between different seasons or years under changing biogeochemical conditions. Unfortunately, however, no benthic Cd fluxes were determined or published prior to our 2017 cruise.

**Referee 2 comment 1:**

*The input data required to verify the authors' flux calculations from concentration gradients between the uppermost pore water sample and the water column sample is absent from the manuscript's figures and tables. That includes the porosity used in equation 3 and the tortuosity calculated therefrom in eq. 2. Nor are the input concentration values given. I cannot glean them from fig. 4, as the scale is too small and most pore water profiles appear to show an enormous spread in the uppermost two data points. What values were used for the calculation? Which one is the true surface in fig. 4? What is the reason for this spread in pore water concentrations so close to the surface - e.g. St1_M136 (Fe: ~ 100 nM and >1 nM) and St4_M136 (Fe: low nM range? and > 6 µM). Comparing bottom water column data (fig. 3) does not suggest one of these plotted pore water values could actually be the bottom water concentration. I think it would be better, to calculate concentration gradients from more than just two data points for Fe and Cd. Perhaps the authors would also consider to show the 0-500 nM range in high resolution, and the few numbers above at lower resolution with a break in the axis between?*

**Response to referee 2 comment 1:**

We fully agree that the input data for the diffusive flux calculations need to be listed in the manuscript. The input data (i.e., porosity, concentration values, in-situ temperature, pressure and salinity) is listed now in the revised version in Tables S1 and S2 in the supplement.

For the diffusive flux calculations we deliberately decided to use the two point concentration gradient between the uppermost pore water sample and the overlying bottom water. We know that this method has its limitations, but need to stay with this approach for the following reasons.

1. This is a commonly used approach which allows us to compare our diffusive fluxes to fluxes from other publications (e.g. Sundby et al., 1986; Warnken et al., 2001; Turetta et al., 2005; Pakhomova et al., 2007; Noffke et al., 2012; Lenstra et al., 2019).

2. The strong gradients between the uppermost pore water and bottom water sample is typically observed within open-marine oxygen minimum zones (e.g. Noffke et al., 2012; Scholz et al., 2019). Since $H_2S$ accumulates close to the surface in these settings, the dissolved Fe peak in pore water is very narrow (1 – 2 cm) and located close to close to the sediment-water interface. Applying more advanced methods for the determination of diffusive fluxes, e.g. curve fitting, would fail to capture the sharp gradients at the sediment surface and, thus, lead to erroneous flux estimates (see also Shibamoto and Harada, 2010; Dale et al., in prep.)

3. Bottom water concentrations are very small (maximum a few hundreds of nM) compared to the concentrations in the uppermost pore water sample (several µM). Therefore, the gradient between pore water and the bottom water is mostly dependent on the pore water concentration. For the same reason, varying the bottom water concentration over the range observed in our data set has a negligible effect on the benthic flux.

We included an explanation for choosing the two point concentration gradient for diffusive flux calculations in lines 321 – 328 in the revised version "We chose to use the commonly applied approach of a two point concentration gradient for the determination of diffusive fluxes, as more advanced curve fitting methods would fail to capture sharp concentration gradients at the sediment surface and, thus, lead to erroneous flux estimates (Shibamoto and Harada, 2010).".

**Referee 2 comment 2:**

*I also don't like the way these gradient-based diffusive flux calculations (lines) are shown over the flux chamber time series data in fig. 5. Better use the water column data as a starting point - since chambers were flushed with ambient bottom water - than the first chamber time series value. There often is quite a difference between the ambient bottom water and the first chamber value that the authors do not explain. Sometimes that difference is hard to understand. In Sta 1 ambient water is higher than the starting value of the chamber (flushed with ambient water). At Sta 9, chamber values (~ 80 nM) are much higher throughout than ambient water (~ 5 nM), but apparently there is no diffusive flux of Fe. That discrepancy does not agree with the authors' general statement in the abstract that the two flux estimates agree with each other.*

**Response to referee 2 comment 2:**

Different Fe concentrations inside and outside of the chamber are to be expected because of diffusive Fe release from the sediment and an accumulation in the enclosed water volume inside the benthic chamber. Furthermore, the bottom water enclosed in the benthic chamber is much closer to the seafloor compared to the bottom water samples collected closest to the seafloor (at 0.5 m). We assume that the concentrations in the bottom water enclosed in the chamber represents the average concentration over the first 30 cm above the seafloor. Outside of the chamber, a concentration gradient between the seafloor and the bottom-near water column is likely to establish because of diffusive release of Fe from the sediment and removal in the bottom water. Therefore, it is not surprising that different concentrations are observed in the chamber and in the bottom water 50 cm above the seafloor.

Furthermore, the water sample at 0.5 m away from the seafloor was collected in sampling bags with peristaltic pumps over a time span of 32 hours. This means that, in contrast to the benthic chamber, where samples were taken within minutes at discrete points in time, short-term concentration changes cannot be seen.

For the reasons given above, we cannot expect to see the same concentration value in different sample types and cannot use bottom water values as a starting point for the incubations.

We added an explanation in lines 432 – 435 in the revised version explaining the different concentrations between bottom water outside and within the benthic chamber "Concentrations of Fe in bottom waters from benthic chamber incubations are mostly higher than in ambient bottom waters because of Fe release from the sediment and an accumulation in the enclosed water volume inside the benthic chamber.".

In the revised version linear regressions for the concentration change over time during the benthic chamber incubation are shown now in Fig. 5 and 7. We prefer to also present the slopes calculated from pore water profiles to facilitate the comparison of benthic fluxes derived by these two different approaches.

**Referee 2 comment 3:**

*Generally, I see an unaddressed problem in that the Fe concentration data for the near-bottom water column and the flux chambers were filtered at 0.2 μm and acidified for later analysis. That would dissolve all colloidal Fe, which I would expect to be quite abundant near the bottom (see e.g. Fitzsimmons et al. 2015, Mar Chem 173 pp 125 for a similar problem in ocean water underneath dust plumes, and Heller et al. 2017, GCA 211, pp 174 for a discussion of particulate - and colloidal - removal of dFe in the Peruvian OMZ). I am aware of the authors citing the latter manuscript, but they do not cite it in the context of colloidal/particulate removal of dFe. I realize, that the necessary ultra-filtration to quantify colloidal Fe was not implemented, and of course, cannot be provided in hindsight. However, I would like to ask the authors to deal with this problem in the discussion. There is so much scatter in the Fe concentration time series from the flux chambers, that I am certain the dFe data suffer from colloids. Colloidal Fe*

*would not count for flux calculations based on pore water - bottom water concentration gradients, and would very much complicate flux estimates based on the linear trend calculation from the flux chamber time series. My quick suggestion would be to provide two estimated for fluxes based on minimum and maximum estimates, but perhaps the authors can come up with a better way to deal with this issue. A discussion of this aspect would really benefit the manuscript. I suggest to discuss this aspect at the end of paragraph 4.1.1. In addition, bioirrigation should contribute considerably to resuspension of particles and colloids as well - a process which the authors briefly mention in lines 441-446 in general terms but do not seem to connect to their scattered flux chamber data.*

**Response to referee 2 comment 3:**

We agree that colloidal or nanoparticulate Fe are likely to play a role in Fe cycling on the Peruvian margin, especially close to seafloor, where particles are quite abundant. Unfortunately, we did not differentiate colloidal and truly dissolved fractions during our sampling, which is why we cannot discuss this aspect based on our data.

The transfer of Fe between dissolved, colloidal and particulate pools is likely to affect the balance between dissolved Fe transport away from the seafloor or re-precipitation and deposition to some extent. To outline the role of suspended particles in modulating benthic Fe fluxes we already refer to the work of Homoky et al., 2012. In addition, we made it clearer in section 4.1.1 that colloidal Fe could modify Fe concentrations within our samples in lines 485 – 493 "Furthermore, colloidal Fe could modify Fe concentrations within our samples and explain some of the fluctuations observed during the incubations. Colloids are quite reactive, they are much more soluble than larger particles and can be rapidly reduced and dissolved in anoxic environments, but they can also aggregate into larger particles (Raiswell and Canfield, 2012). The transfer of Fe between dissolved, colloidal and particulate pools is likely to affect the balance between Fe transport and re-precipitation and -deposition to some extent. However, since did not differentiate between colloidal and truly dissolved fractions during our sampling, we cannot discuss this aspect further based on our data.".

We don't understand the reference to Heller et al., 2017 in the context of colloids as this publication does not discuss the role of colloidal Fe.

Finally, we fully agree with the reviewer's comment that further research on particle dissolved interactions is needed and we mention this in lines 544 – 546 in the revised version "Further research on dissolved-particulate interactions, including the role of colloidal Fe, in bottom waters is needed to better constrain how sedimentary Fe fluxes are modified in the near-bottom water column.".

We do connect our scattered flux chamber data at stations 1, 9 and 10 to bioturbation and bioirrigation in lines 433 – 436. We added that bioturbation and bioirrigation could also lead to particle or colloid resuspension in lines 483 – 484 in the revised version "Bioturbation and bioirrigation could also contribute to particle resuspension at oxic stations, thus leading to removal of dissolved Fe.".

**Referee 2 comment 4:**

*In the abstract, lines 20-23, I would suggest the authors phrase the suggested "agreement" of different flux calculations somewhat more careful. This is not a very convincing statement even without considering the colloid problem. At best, there is agreement for some of the stations.*

**Response to referee 2 comment 4:**

Following the reviewer's recommendation, we rephrased lines 20 – 23 in the revised version "Diffusive Fe fluxes and Fe fluxes from benthic chamber incubations (-0.3 – -17.5 mmol m-2 y-1) are broadly consistent at most stations, indicating that diffusion is the main transport mechanism of dissolved Fe across the sediment-water interface. ".

**Referee 2 comment 5:**

*In the conclusion, I don't think it is appropriate to claim that Fe just diffuses out of the sediment column. The authors themselves have made the case for biorrigation and bioturbation as a significant process for Fe enrichment in the flux chambers.*

**Response to referee 2 comment 5:**

We rephrased this part of the conclusions and only refer to diffusion as being the dominant process for Fe fluxes within the permanent oxygen minimum zone, where bottom-dwelling macrofauna is absent in lines 742 – 743 in the revised version "Within the OMZ, where bottom dwelling macrofauna is absent, diffusion is the main process that transports Fe from the sediment pore water into the bottom water".

We are aware that additional processes could play a role within the permanent oxygen minimum zone. However, we feel it is appropriate to only refer to the main process in the conclusions.

**Technical comments referee 2:**

*Technical aspects that should be corrected: a) Please show cruise AND station number in figures 1 and 2. b) I believe it would be useful to show the H2S data along with the Cd data in fig. 6, since they are discussed together.*

**Response to technical comments referee 2:**

We implemented your suggestions. Station numbers were added to the map in Fig. 1 and to the caption of Fig. 2. Hydrogen sulfide concentrations in pore waters are also plotted in Fig. 6 now.

[revised manuscript text omitted]

**Supplement**

Table S1: Input data from M136 & M137 for diffusive flux calculations.

| station | Fe(II) concentration bottom water | Fe(II) concentration in pore water at sediment surface | Cd concentration bottom water | Cd concentration in pore water at sediment surface | porosity | temperature | pressure | salinity |
|---|---|---|---|---|---|---|---|---|
| | (µM) | (µM) | (nM) | (nM) | | (C°) | (bar) | |
| 1 | 0.10 | 1.06 | 0.22 | 0.90 | 0.93 | 16.17 | 8.77 | 35.06 |
| 3 | 0.17 | 0.47 | 0.81 | 0.65 | 0.95 | 13.96 | 13.90 | 34.97 |
| 4 | 0.00 | 6.34 | 0.66 | 0.47 | 0.96 | 13.96 | 15.40 | 34.97 |
| 5 | 0.00 | 0.59 | 0.68 | 0.44 | 0.96 | 13.21 | 20.40 | 34.92 |
| 6 | 0.06 | 3.14 | 0.76 | 0.55 | 0.95 | 13.33 | 25.60 | 34.94 |
| 9 | 0.00 | 0.00 | 0.99 | 1.25 | 0.74 | 6.28 | 75.00 | 34.55 |
| 10 | 0.01 | 0.37 | - | - | 0.61 | 4.36 | 98.20 | 34.55 |

Table S2: Input data from M92 for diffusive flux calculations.

| station | Fe(II) concentration in bottom water | Fe(II) concentration in pore water sediment surface | porosity | temperature | pressure | salinity |
|---|---|---|---|---|---|---|
| | (µM) | (µM) | | (C°) | (bar) | |
| 1 | 0.70 | 4.83 | 0.96 | 13.99 | 8.1 | 34.98 |
| 3 | 0.21 | 0.77 | 0.98 | 13.84 | 13.90 | 34.98 |
| 4 | 1.15 | 1.80 | 0.96 | 13.77 | 15.50 | 34.96 |
| 5 | 0.03 | 0.31 | 0.96 | 13.16 | 20.50 | 34.94 |
| 6 | 0.20 | 2.13 | 0.96 | 13.16 | 25.40 | 34.94 |
| 9 | 0.03 | 0.03 | 0.84 | 5.17 | 78.30 | 34.54 |
| 10 | 0.00 | 0.06 | 0.74 | 4.62 | 103.4 | 34.55 |

Table S3: Silicic acid (Si(OH)$_4$) concentrations in the near-bottom water column.

| | Station 1 | Station 3 | Station 4 | Station 5 | Station 6 | Station 9 |
|---|---|---|---|---|---|---|
| distance from seafloor | Si(OH)$_4$ concentration | Si(OH)$_4$ concentration | Si(OH)$_4$ concentration | Si(OH)$_4$ concentration | Si(OH)$_4$ concentration | Si(OH)$_4$ concentration |
| (m) | (μM) | (μM) | (μM) | (μM) | (μM) | (μM) |
| 0.5 | 31.28 | 29.74 | 24.86 | 24.32 | 27.28 | - |
| 1 | 31.63 | 29.22 | 24.51 | 24.20 | 27.14 | 56.22 |
| 2 | - | 29.2 | 23.61 | 24.35 | 27.02 | 56.04 |
| 3 | 31.38 | 28.66 | 24.36 | - | - | 55.96 |
| 4 | 31.27 | 28.17 | 24.20 | 24.00 | 27.04 | 55.44 |

Table S4: Equations for linear regressions from benthic chamber incubation data (Y =
concentration, X = incubation time) and coefficients of determination (R$^2$).

| station | Fe linear regression | Fe R$^2$ | Cd linear regression | Cd R$^2$ |
|---|---|---|---|---|
| 1 | Y = 0.88 * X + 186.18 | 0.01 | Y= 0.00 * X + 0.068 | 0.06 |
| 4 | Y = 4.78 * X + 7.07 | 0.72 | Y = -0.01 * X + 0.55 | 0.96 |
| 5 | Y = -0.89 * X + 89.28 | 0.1 | Y = -0.01 * X + 0.53 | 0.89 |
| 6 | Y = 2.32 * X + 57.34 | 0.47 | Y = -0.01 * X + 0.58 | 0.87 |
| 9 | Y = 2.95 * X + 76.38 | 0.49 | Y = 0.00 * X + 1.01 | 0.36 |
| 10 | Y = 0.67 * X + 28.20 | 0.12 | Y = 0.00 * X + 0.99 | 0.07 |

---

## Author Response (AR2)

In this response letter the *referee comments are listed in blue italic font*, followed by the authors response.

*Dear colleagues,*

*The efforts made to introduce more nuance to their statements regarding uncertainties of flux calculations from different methods used here is appreciable. However, I think some more work should be done.*

*For instance, suggestions made by Michael Staubwasser to show better pore water profiles by zooming around the interface and try to fit more than two points are crucial here. The arguments advanced to stick on 2 points modelling are, in my opinion, not really convincing.*

*argument 1. I am not sure that saying "many others did, we should do" is very scientific.*

*In addition, looking back to papers from Sundby and collaborators (Sundby et al, 1986 and Westerlund et al 1986), it seems that they do not calculate fluxes out of porewaters. In the second one, they compare overlying water and the first layer of porewater to predict flux direction, they do not go further.*

*argument 2*

*the authors say that "advanced curve fitting methods would fail to capture sharp gradients..." in order to justify why they do a simple 2 points fitting to calculate diffusive flux. I agree that there is no problem using linear regressions with pour-resolved vertical profiles. The problem here is that between a 2-points and a 3-points regression, there is a big gap in terms of modelling validity.*

*argument 3*

*Indeed, if bottom water concentration is low so it does not affect the gradient, however the first porewater concentration does. The latter concentration is the result of a certain volume of porewater mixed altogether during sampling and extraction. We can imagine that some of it had oxygen, some of it sulphide and the final concentration is the result of thermodynamic equilibium, not only due to mixing porewaters of different concentrations.*

*This is better said by Warnken et al., 2001 as follows:*

*"The diffusive fluxes of Fe agreed with measured fluxes at station 1, but diffusive flux estimates of Fe at stations 2 and 3 were greater than the measured chamber fluxes, possibly indicating that pore water Fe concentration gradients did not accurately represent actual interfacial concentration gradients at these stations. This could be due to less steep gradients in the upper mm of sediment suggesting that the 1 cm sampling resolution used here was not adequate. Accounting for the reduced slope very near*

*the sediment–water interface would result in lower calculated diffusive fluxes that would possibly agree more closely with measured fluxes. Additionally, rapid oxidative processes at or near the sediment–water interface could also reduce the amount of Fe released from the sediments, as concentration increases during the incubation experiments conducted at stations 2 and 3 were reduced when compared to station 1."*

*Having a third point, helps to show that the trend is maintained despite interface processes.*

*I would suggests the authors to simply explain that linear regressions were made because of scarcity of data due to sharpness of gradients at a centimetre resolution profiling. In addition, I would put a warning about the precision of numbers obtained.*

We would like to thank the reviewer for providing us with additional referee comments and we appreciate the reviewer's remaining concerns regarding methodology.

He rightly pointed out that a three-point regression would be preferable for the calculation of benthic diffusive fluxes because of a lower statistical uncertainty. He also acknowledged problems related to the coarse vertical resolution of our pore water profiles and the sharpness of the concentration gradients within the surface sediment.

Due to the soft texture of the sediment, it was not possible to sample the pore water at a higher resolution within the first centimeter of the sediment. Furthermore, as stated in the first response to the reviewer, Fe concentrations in pore water drop sharply shortly below the sediment surface because of high rates of bacterial sulfate reduction and Fe sulfide precipitation within the surface sediment. Consequently, choosing an additional concentration point, that would be located below this drop in concentration, would simply lead to artificially low fluxes. Although, the uncertainty of a three-point approach would be lower from a statistical point of view, it would clearly lead to erroneous results. To illustrate this and following the reviewers' recommendation, we added close-up profiles of pore water concentrations within the upper 5 cm of the surface sediment to the supplement (Fig. S1 and S2). Furthermore, we added a more detailed explanation, including the reviewers' suggestions, to state why we applied the two-point gradient in lines 323 – 330 in the revised manuscript "Due to the coarse resolution of our pore water profiles and the steep gradients between the uppermost pore water and bottom water sample (see close-up profiles, Fig. S1 and S2, in the supplement), we chose to follow previous studies (e.g., Noffke et al., 2012; Lenstra et al., 2019; Scholz et al., 2019) and calculate diffusive benthic fluxes based on a two point concentration gradient. Including deeper samples into a linear regression or applying more advanced curve fitting methods would reduce the statistical uncertainty but fail to capture the sharp concentration gradients at the sediment surface, thus, leading to erroneous flux estimates (cf. Shibamoto and Harada, 2010). " We hope that this will clarify, why we have to choose the applied method. A more detailed discussion of the early diagenetic processes leading to steep and variable gradients within the surface sediment (bacterial sulfate reduction, precipitation of Fe monosulfides, etc.) would be inappropriate in the method section.

*Another point that was not fully addressed is the uncertainties on incubation calculations. You must have errors over 100 % with a R2 of 0.5. You should look in detail what are uncertainties and which regression has statistical significance.*

Following the reviewer's recommendation, we calculated the uncertainty of our in-situ benthic fluxes by propagating the errors of the linear regressions of the concentration change over time (see Tables 2 and 3 in the revised version). Furthermore, following previous studies (e.g., Friedrich et al. (2002); Lenstra et al. (2019)), only fluxes with a linear regressions with a $R^2 > 0.3$ are reported in Tables 2 and 3 (See lines 338 - 339 in the revised manuscript).

*All this should lead to a discussion about how we can do better. As it is, it gives me the idea that since 1986 no analytical progress was made to improve spatial resolution and to limit incubation time and increase incubation replicates. Maybe that is the case, but it has to be said somewhere*

We agree that the advantages and disadvantages of the various methods and approaches for the determination of benthic fluxes is an exciting topic. However, the focus of our study is the benthic biogeochemical cycling of Fe and Cd in oxygen minimum zones and not the technical aspects of the methods applied. Therefore, we feel that providing an extended discussion about technical limitations and potential improvements to the methods is beyond the scope of our work.

All the methods and approaches applied in our study are commonly applied in benthic biogeochemical studies by various research groups around the world. We agree that improvements to these methods are possible. On the other hand, for a comparison of benthic fluxes from different locations or seasons (note that we compare our Fe flux data to those determined in the same area a couple of years earlier), it is quite useful to stick to methodologies that have been applied before. For this reason, we refer to previous studies in the description of our methodology.

[revised manuscript text omitted]

---

## Author Response (AR3)

We appreciate the reviewer's comments and did our best to address them. Our response and modifications to the manuscript are appended below (*Reviewer comments in blue italic*/response in regular font):

*The point about best not to do flux calculations between just two points was not made - as the authors' appear to argue about - out of concerns regarding statistical precision, but of accuracy. Even if the pore water profile only permits to use one data point, a two-point line approach is never satisfactory. It does not look to me like the authors followed the editor's simple and well meaning suggestion of a way forward.*

The reviewer criticized again that we follow a two-point approach to calculate diffusive benthic fluxes across the sediment-water interface and recommends instead to interpolate between multiple points to derive the concentration gradient. We are unaware of any specific suggestions from the editor regarding the best way to proceed with this point of criticism and it is clearly not our goal to recklessly decline well-intentioned comments from reviewers or editors.

While we would love to follow the reviewer's recommendation (to add an additional point to our diffusive flux calculations) in order to get our study accepted, we cannot do this because following this approach would yield benthic flux estimates that are incorrect. We would like to explain this and defend our view with a sketch illustrating the scientific rationale behind our flux calculations (see Figure 1 below). Peru margin sediments are characterized by high iron concentrations within the surface sediments (~1 cm) but the concentrations decline shortly below because dissolved iron is precipitated with hydrogen sulfide that is generated by bacterial sulfate reduction (Scholz et al., 2011; Noffke et al., 2012). Therefore, dissolved iron is transported by diffusion both in an upward direction across the sediment-water interface and in a downward direction into the sulfidic zone of the sediments. Lumping samples from both the upward- and downward-directed concentration gradients (light and dark gray arrays in Figure 1) into one single flux calculation may be statistically desirable but is mechanistically wrong.

[Figure]

*Figure 1. Sketch illustrating the scientific rationale of benthic flux calculations.*

The accuracy of flux calculations based on curve-fitting exercise depends on the choice of the fitting function and the precision of the measured data. In the ideal case, a perfect fit function that considers each source and sink would pass through all the measured data points, and result in the same flux that we had calculated. Unfortunately, there is no single fitting function for dissolved iron that we are aware of. The largest uncertainty in our calculations lies not with the goodness of fit below the upper centimeters (which plays no role in the driving the Fe flux across the sediment surface) but in the resolution of the measured data close to the sediment-water interface. The only way to get a statistically better estimate of the dissolved iron flux across the sediment-water interface is to collect multiple samples within the uppermost 1 cm of the sediment. Given the fluffy character of surface sediments on the Peruvian margin, such an approach is not feasible using conventional approaches for pore water recovery. The other reviewer noted that more elaborate sampling techniques are available. That is true and we added a paragraph to the manuscript (lines 322 – 330) noting that those methods were not available during our research expedition " Due to the coarse resolution of our pore water profiles and the steep gradients between the uppermost pore water and bottom water sample (see close-up profiles, Fig. S1 and S2 in the supplement), we chose to follow previous studies (Pakhomova et al., 2007; Noffke et al., 2012; Scholz et al., 2016, 2019; Lenstra et al., 2019) and calculate diffusive benthic fluxes based on a two point concentration gradient. Including deeper samples into a linear regression or applying more advanced curve fitting methods would reduce the statistical uncertainty but fail to capture the sharp concentration gradients at the sediment surface and thus lead to erroneous flux estimates (cf. Shibamoto and Harada, 2010)." We note, however, that those methods are typically applied in coastal research and not utilized on a routine basis on sea-going expeditions during which multiple cores are collected within a short time period. Moreover, our group (Scholz et al., 2011, 2016, 2019; Noffke et al., 2012; Dale et al., 2015) and multiple other research groups around the world (Sundby et al., 1986; Warnken et al., 2001; Pakhomova et al., 2007; Covelli et al., 2008; Shibamoto and Harada, 2010; Lenstra et al., 2019) have been following this approach of using two concentration points to calculate benthic fluxes for years demonstrating that it is widely accepted by the scientific community.

We discuss and acknowledge all the shortcomings of our approach in the manuscript and expanded on this aspect even further in the revised version. However, there is no scientifically sound alternative to this approach and we sincerely hope that we could clarify this with our letter.

*Between line 393 and 396 the authors claim that the two methods of flux calculations are "largely consistent in direction and slope". I still don't see that. There is some agreement, but almost as much disagreement. Table 2 has three entries where both methods' numbers are given. Two agree in direction, one doesn't. Only one of the two is reasonably close regarding the slope, the other is a factor of two different. In Fig. 5, we have got the two matches in direction (Sta 4 and 6), but three mismatches (Sta 5 and 9 are going in opposite directions, 10 shows no flux vs outward flux; with Sta 1 I don't understand, why the benthic flux line does not go through the data). I cannot help, but to me, the claim that there "largely" is an agreement looks a bit preconceived. My conclusion is, that either the methodology is limitied under the given circumstances, or there are internal dynamics in the Fe cycle that are not revealed by the study.*

The reviewer correctly mentioned that there is not an agreement in direction and slope between diffusive fluxes and fluxes from benthic chamber incubations at all stations. To make clear that the agreement, between the fluxes derived from the two different methods, is in particular the case for Station 4 and 6, located inside the OMZ, we realized the following modifications to the manuscript.

Lines 20 – 24 "Diffusive Fe fluxes and Fe fluxes from benthic chamber incubations (-0.3 – -17.5 mmol $m^{-2}$ $y^{-1}$) are broadly consistent at stations within the oxygen minimum zone, where the flux magnitude is highest, indicating that diffusion is the main transport mechanism of dissolved Fe across the sediment-water interface."

Lines 393 – 397 "At some stations the incubation data were consistent in direction and slope with the diffusive fluxes. In particular at Station 4 and 6 inside the OMZ, where the highest diffusive fluxes of -17.5 and -8.0 mmol $m^{-2}$ $y^{-1}$ were observed, expected and observed concentration gradients were in good agreement"

Lines 440 – 444 "Consistent with this notion, the slope calculated from benthic diffusive fluxes is in good agreement with the concentration gradients observed within the benthic chambers at two stations within the OMZ (Station 4 and 6) (Fig. 5). Moreover, our fluxes from benthic chamber incubations and diffusive fluxes are of similar magnitude at these stations (Table 2)."

Regarding the reviewer's concerns that either the methodology is limited or that there are internal dynamics in the Fe cycle, that are not revealed by our study, leading to the variations between the two flux estimates we can refer to chapter 4.1.1 in our manuscript. In this chapter we discuss different aspects that can exert an important control on the benthic Fe cycle, in particular during benthic chamber incubation, and that can modify Fe fluxes and thus leading to variations between the two flux estimates. Further, in lines 447 – 451 we mention that scatter in data from benthic chamber incubations is a common observation "Some of the concentration gradients in benthic chambers are non-linear, indicating that the Fe flux was not constant during the incubations. This is a common observation in Fe flux data from benthic chamber incubations and higher Fe fluxes generally have higher $R^2$ values for the linear regressions (Friedrich et al., 2002; Turetta et al., 2005; Severmann et al., 2010; Lenstra et al., 2019)." This may be a potential limitation of the method when fluxes are low and lead to mismatches between different flux estimates (Station 1, 9 and 10). However, the similarity of benthic fluxes within the oxygen minimum zone (Station 4 and 6) still supports our conclusion, that diffusion is the main transport mechanisms of dissolved Fe from pore water into incubated bottom water. At the same time, the agreement in flux magnitude between the two different methods, also suggests that our diffusive flux estimates from a two-point concentration gradient are correct.

[revised manuscript text omitted]